# Potassium channel-based optogenetic silencing

Yinth Andrea Bernal Sierra[1], Benjamin R. Rost[2,3], Martin Pofahl[4], António Miguel Fernandes[5], Ramona A. Kopton[6,7], Sylvain Moser[5,8], Dominik Holtkamp[4], Nicola Masala[4], Prateep Beed[2,9], John J. Tukker [2,3], Silvia Oldani[2,3], Wolfgang Bönigk[10], Peter Kohl[6], Herwig Baier[5], Franziska Schneider-Warme[6], Peter Hegemann[1], Heinz Beck[4], Reinhard Seifert[10] & Dietmar Schmitz[2,3,9,11,12,13]

Optogenetics enables manipulation of biological processes with light at high spatio-temporal resolution to control the behavior of cells, networks, or even whole animals. In contrast to the performance of excitatory rhodopsins, the effectiveness of inhibitory optogenetic tools is still insufficient. Here we report a two-component optical silencer system comprising photo-activated adenylyl cyclases (PACs) and the small cyclic nucleotide-gated potassium channel SthK. Activation of this 'PAC-K' silencer by brief pulses of low-intensity blue light causes robust and reversible silencing of cardiomyocyte excitation and neuronal firing. In vivo expression of PAC-K in mouse and zebrafish neurons is well tolerated, where blue light inhibits neuronal activity and blocks motor responses. In combination with red-light absorbing channelrhodopsins, the distinct action spectra of PACs allow independent bimodal control of neuronal activity. PAC-K represents a reliable optogenetic silencer with intrinsic amplification for sustained potassium-mediated hyperpolarization, conferring high operational light sensitivity to the cells of interest.

[1] Institute of Biology, Experimental Biophysics, Humboldt-Universität zu Berlin, Invalidenstr. 42, 10115 Berlin, Germany. [2] Charité-Universitätsmedizin Berlin, Corporate member of Freie Universität Berlin, Humboldt-Universität zu Berlin, and Berlin Institute of Health, Neuroscience Research Center, 10117 Berlin, Germany. [3] German Center for Neurodegenerative Diseases (DZNE), 10117 Berlin, Germany. [4] Institute for Experimental Epileptology and Cognition Research, University of Bonn Medical Center, Sigmund-Freud Str. 25, 53127 Bonn, Germany. [5] Department Genes–Circuits–Behavior, Max Planck Institute of Neurobiology, Am Klopferspitz 18, 82152 Martinsried, Germany. [6] Institute for Experimental Cardiovascular Medicine, University Heart Center, Medical Center – University of Freiburg, and Faculty of Medicine, University of Freiburg, Elsässer Str. 2Q, 79110 Freiburg, Germany. [7] Faculty of Biology, University of Freiburg, Schänzlestr. 1, 79104 Freiburg, Germany. [8] International Max Planck Research School for Translational Psychiatry (IMPRS-TP), 80804 Munich, Germany. [9] Berlin Institute of Health (BIH), 10178 Berlin, Germany. [10] Molecular Sensory Systems, Center of Advanced European Studies and Research (caesar), Ludwig-Erhard-Allee 2, 53175 Bonn, Germany. [11] Bernstein Center for Computational Neuroscience, 10115 Berlin, Germany. [12] Cluster of Excellence NeuroCure, 10117 Berlin, Germany. [13] Einstein Center for Neurosciences Berlin, 10117 Berlin, Germany. These author contributed equally: Yinth-Andrea Bernal-Sierra, Benjamin R. Rost, Martin Pofahl. Correspondence and requests for materials should be addressed to R.S. (email: Reinhard.Seifert@caesar.de) or to D.S. (email: Dietmar.Schmitz@charite.de)

Reversible light-based silencing is a key requirement for investigations into the role of genetically defined cell populations, for example to explore effects of individual neurons in controlling network activity and behavior[1], or to investigate the effects of local conduction block zones in the heart[2]. Current optogenetic inhibitors comprise inward-directed chloride pumps, outward-directed proton or sodium pumps, or chloride-conducting channelrhodopsins[3–9]. Their application, however, can suffer from paradoxical effects due to unintended changes in ion distributions[10–13], and most of them require high expression levels and persistent high-intensity illumination for efficient inhibition of cell activity. The generation of a light-activated potassium ($K^+$) channel has the potential to overcome many of these limitations, primarily because $K^+$ conductances underlie the resting state of excitable cells, so their activation can generate strong inhibition without major changes in transmembrane ion gradients. However, efforts to generate a widely usable light-gated $K^+$ conductance have not been successful so far. Available synthetic light-gated $K^+$ channels suffer from poor expression in mammalian cells[14], the requirement for additional cofactors[15–17], irreversibility of activation[18], or the large size of the encoding genes[19], all of which have hampered their widespread application.

In order to establish a light-controlled $K^+$-based hyperpolarization tool, we conceived an optogenetic system consisting of a cyclic nucleotide-gated (CNG) $K^+$ channel and a photo-activated nucleotidyl cyclase (PAC). This concept requires a cyclic nucleotide-gated channel with high preference for $K^+$ over $Na^+$ and $H^+$ that shows high functional expression levels in mammalian cells. Classic mammalian CNG channels or hyperpolarization-activated and cyclic nucleotide-gated (HCN) channels do not show a high $K^+$ selectivity over $Na^+$ and $H^{+20,21}$. A cGMP-gated $K^+$ channel was found in *Arbacia punctulata* sperm[22], but this channel bears a pseudotetrameric architecture and is therefore too large for applications relying on viral gene transfer. For the prokaryotic $K^+$ channels MLoK1 from *Mesorhizobium loti*[23] or MmaK from *Magnetospirillum magnetotacticum*[24] evidence for functional expression in mammalian cells is lacking[25]. In contrast, the small cAMP-gated $K^+$ channel SthK from *Spirochaeta thermophila* is well expressed in mammalian cells, and it carries an exceptionally large single channel current ensuring robust repolarization or hyperpolarization[26–28], rendering it the most promising candidate for a silencer system.

PACs are blue-light-activated cyclases consisting of a blue light receptor using flavin (BLUF) domain coupled via a linker to the cyclase domain. They were identified in various species, providing a range of light sensitivities and enzyme kinetics[29]. The first-discovered PACα and PACβ from Euglenoid algae[30] are large proteins of 105 and 90 kDa, respectively, and only poorly soluble. In contrast, the small bPAC (33 kDa) of the soil bacterium *Beggiatoa* and similar small PACs from other species can be well expressed in a variety of host cells[29,31–33].

We characterized various combinations of the SthK channel with small PACs in cell lines to obtain robust light-activated $K^+$ currents. Among these, the combinations of bPAC or TpPAC with SthK showed most promising results, and were further validated for general applicability in excitable cells, including cardiomyocytes and different types of neurons. The 'PAC-K' silencer efficiently suppresses action potential (AP) generation and contraction in cardiomyocytes, and neuronal firing both in vitro and in vivo. We found that the combination of bPAC and SthK yields a highly light sensitive $K^+$-based hyperpolarizer that requires minimal illumination for long lasting, fully reversible hyperpolarization, thereby providing an exceptionally efficient optogenetic silencer. Importantly, the $K^+$ currents induced by PAC-K activation mimic nature's own repolarization mechanism

for resting cell polarization, which is energetically favorable and not associated with major alterations in other ion concentration gradients.

## Results

**Engineering of PAC-K.** We tested two single-vector coexpression strategies for PACs and SthK (Fig. 1a): either with the PAC separated from the channel by a self-cleaving 2A peptide at the SthK C-terminus (split-PAC-K), or with the PAC N-terminally fused to SthK (fused-PAC-K). The design of the split system aimed to exclude any impairment of SthK function by the attachment of the PAC, while the fused system should achieve very short cAMP diffusion distances between enzyme and channel. In ND7/23 cells expressing split-bPAC-K with bPAC from *Beggiatoa*[31], a 10-ms light pulse evoked intensity-dependent outward currents that lasted for several seconds (Fig. 1b, c; Supplementary Fig. 1a). At 50% light saturation ($EC_{50}$: 0.86 mW $mm^{-2}$), currents reached a maximum within ~12 s (Supplementary Fig. 1a), and decayed exponentially with $\tau_{off} = 33 \pm 17$ s (median ± standard deviation). To achieve faster off-kinetics we tested less active PACs from *Naegleria gruberi* (NgPAC1)[34], *Turneriella parva* (TpPAC)[35], *Oscillatoria acuminate* (OaPAC)[36], and *Ilumatobacter coccineus* (IcPAC) (Fig. 1b, c). Among these, TpPAC displayed four times faster off-kinetics and a 10 times reduced light sensitivity (Fig. 1c, e). Comparison of various split-PAC-K versions with the widely used light-driven $H^+$ pump Arch3[3,4] showed that only split-bPAC-K provided a higher effective light sensitivity at $EC_{50}$ than Arch3, whereas other PAC-expressing cells were less sensitive (Supplementary Fig. 1b). We surmise that this result might be due to a maximal activation of SthK at light intensities not saturating bPAC, because photon capture of flavins is lower compared to retinal, due to a smaller extinction coefficient[37]. However, all split-PAC-K versions transported considerably more charges per absorbed photon than Arch3 (up to a factor of $3 \times 10^4$) due to the long-lasting activation of currents (Fig. 1c; Supplementary Fig. 1b). We simultaneously monitored the photoactivated $K^+$ current and intracellular cAMP levels using a red fluorescent cAMP reporter[38] (Fig. 1d). Current activation slightly trailed the cAMP increase ($\tau_{current}/\tau_{cAMP} = 1.52 \pm 1.26$, $n = 20$ from seven cells) while current deactivation preceded the decline in cAMP ($\tau_{current}/\tau_{cAMP} = 0.33 \pm 0.22$, $n = 9$ from three cells). Comparing the fused-PAC-K and split-PAC-K strategies for both bPAC and TpPAC revealed similar cellular light sensitivities and current kinetics for both PAC-K systems, with smaller photocurrents for fused-bPAC-K (Fig. 1e–h), and poor photocurrents and onset kinetics for fused-TpPAC-K compared to split-TpPAC-K (Fig. 1f, g). We also assessed SthK activation by the bPAC-S27A variant with 15-nm red-shifted absorbance and further reduced dark activity[39], but found no difference compared to activation by bPAC (Supplementary Fig. 1d–g). In summary, both fused-PAC-K and split-PAC-K systems were identified as promising silencing tools.

**Split-bPAC-K silences cardiomyocyte activity.** To test the bPAC-K system in excitable cells, we first transduced rabbit primary ventricular cardiomyocytes with adenovirus for fused-bPAC-K or split-bPAC-K. While the fusion construct failed to elicit any light-induced response, split-bPAC-K-expressing myocytes (Fig. 2a) showed robust outward currents following short blue light exposure (10 ms, 460 nm). Amplitude and duration of currents were graded with light intensity (Fig. 2b–d). Single light pulses suppressed electrically evoked APs for extended time periods ($283 \pm 43$ s at 4 mW $mm^{-2}$, Fig. 2e), with the duration of the AP-free interval scaling linearly with the $K^+$ current duration

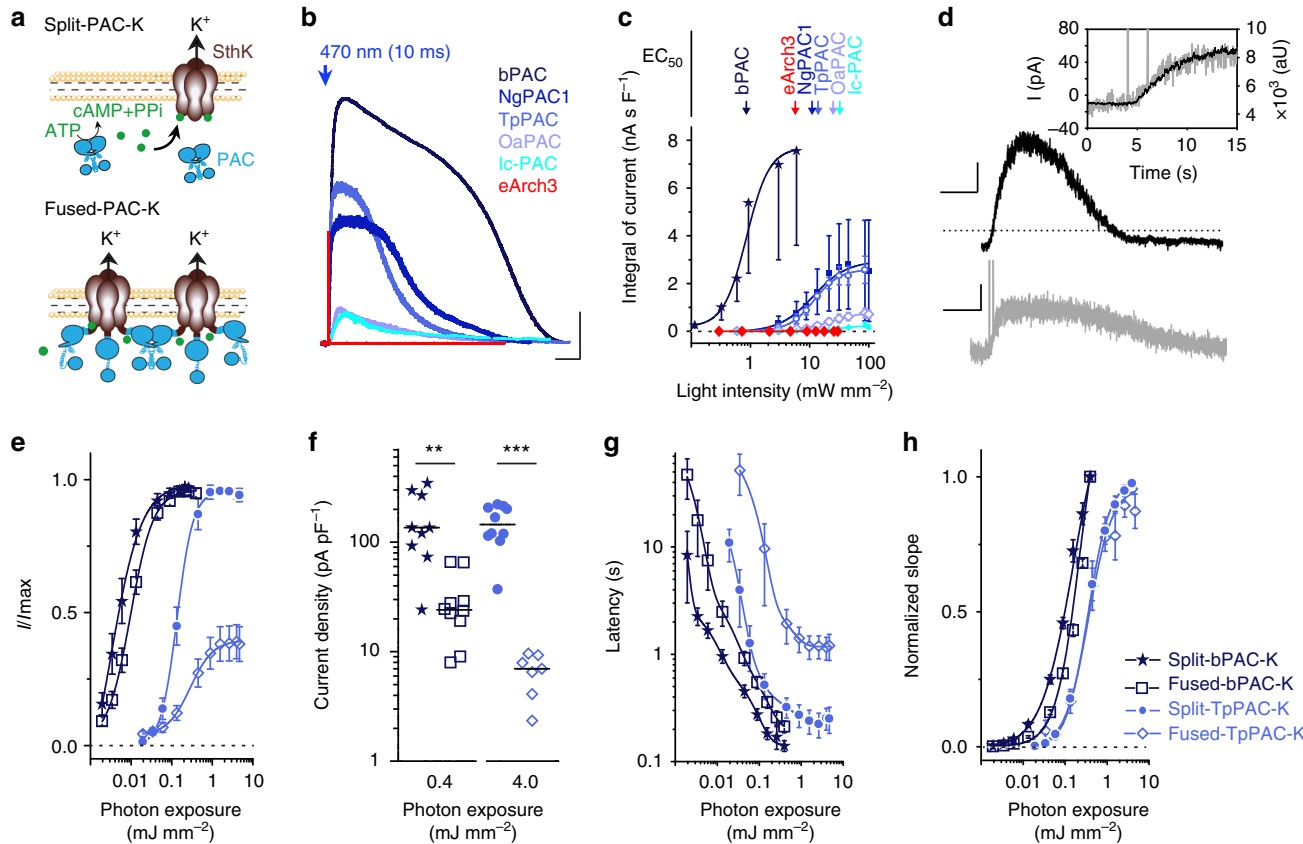

**Fig. 1** Optogenetic activation of SthK channels by various PACs in cell lines. **a** Illustration of the split-PAC-K and fused-PAC-K construct design for SthK co-expression with PACs. **b** Photocurrents elicited by different split-PAC-K variants after 10 ms exposure to a 470 nm light pulse. Scale bars: 10 s, 0.5 nA. **c** Normalized integrals of photocurrents as a function of the intensity of a 10 ms light pulse. Arrows denote the $EC_{50}$ for current activation by light ($n = 3$ cells, two cultures). **d** Combined whole-cell patch-clamp (black) and optical recording (gray) of a HEK 293 cell expressing fused-bPAC-K and a fluorescent cAMP sensor. bPAC was excited with two 1 ms light flashes (470 nm). The dotted line indicates the zero current level. Scale bars: 20 s, 20 pA (upper trace); 20 s, 2000 aU (lower trace). **e** Comparison of photocurrent amplitudes generated by bPAC or TpPAC and SthK as split or fused constructs. **f** Current density comparison at saturating photon exposures ($^{**}p < 0.01$; $^{***}p = 0.0001$, Wilcoxon rank sum test). **g**, **h** Latency and photocurrent rise upon excitation with blue light at different photon densities. In **e–h**, bPAC and TpPAC were activated for 10 and 100 ms, respectively (split-bPAC-K: $n = 9$, four cultures; fused-bPAC-K and split-TpPAC-K: $n = 10$, three cultures; fused-TpPAC-K: $n = 8$, five cultures). Error bars in all graphs represent SD

(Fig. 2f). AP duration at 90% repolarization ($APD_{90}$) recovered to baseline values within 10 s after reoccurrence of excitability, and importantly, was not different from $APD_{90}$ measured in non-transduced control cells (Fig. 2g). Light-activation of split-bPAC-K also effectively silenced cardiomyocyte contractions (Fig. 2h). A complex illumination protocol (Fig. 2i) revealed a direct dependence of the contraction-free interval on light-pulse duration (Fig. 2j), and highly reproducible contraction-free intervals that are independent of previous inhibitory events (Fig. 2k).

**PAC-K characterization in neurons**. Next we assessed the expression of both PAC-K variants in cultured hippocampal neurons. The small size of the PAC-K-encoding DNA sequences allowed gene delivery by a single adeno-associated virus (AAV). Expression of either fused-bPAC-K or split-bPAC-K did not affect intrinsic electrophysiological membrane properties (Supplementary Table 1). In cells expressing split-bPAC-K, 100 ms pulses of 385 or 470 nm light with >1 mW mm$^{-2}$ shifted the minimal current necessary to evoke an AP (rheobase) by more than 600 pA and suppressed spiking by hyperpolarization for up to 1 min (Fig. 3a–c). SthK-mediated changes in membrane voltage were in accordance with the reversal potential of K$^+$ (Fig. 3d). The underlying outward currents depended on the intensity of the applied light pulse (Supplementary Fig. 2a–e),

with peak amplitudes of $861 \pm 119$ pA and maximum duration of $81 \pm 9$ s. Importantly, split-bPAC-K functioned as an integrator of the activation light. When increasing the light pulse duration from 100 ms to several seconds, we were able to elicit sustained hyperpolarizing currents with light intensities as low as 0.4 or 0.04 mW mm$^{-2}$ (Fig. 3e). Increasing the ambient temperature from 20 to 35 °C shortened the current rise-time and duration, but did not change the amplitude (Fig. 3f). The hyperpolarization was specifically mediated by activation of the SthK channel, as blue light-exposure of cells expressing only bPAC caused small, depolarizing currents of $-36.1 \pm 7.4$ pA (Supplementary Fig. 2f, g). Split-TpPAC-K also elicited hyperpolarizing currents in neurons, but required 10× higher light intensities for efficient activation compared to split-bPAC-K (Supplementary Fig. 3a). Split-TpPAC-K-mediated currents were four times faster compared to split-bPAC-K, but showed a more pronounced run-down during repetitive activation (Supplementary Fig. 3b–e).

Since bPAC is specifically activated at wavelengths <500 nm (Supplementary Fig. 1g), we tested whether one can optically elicit independent excitation and inhibition in cultured neurons coexpressing split-bPAC-K and the red-shifted channelrhodopsin bReaChES[40]. In whole-cell current-clamp mode we repetitively triggered APs by bReaChES activation with 550 nm light pulses. Single 100 ms pulses of 385 or 470 nm light abolished bReaChES-driven spiking in an intensity-dependent manner (Fig. 4a–c).

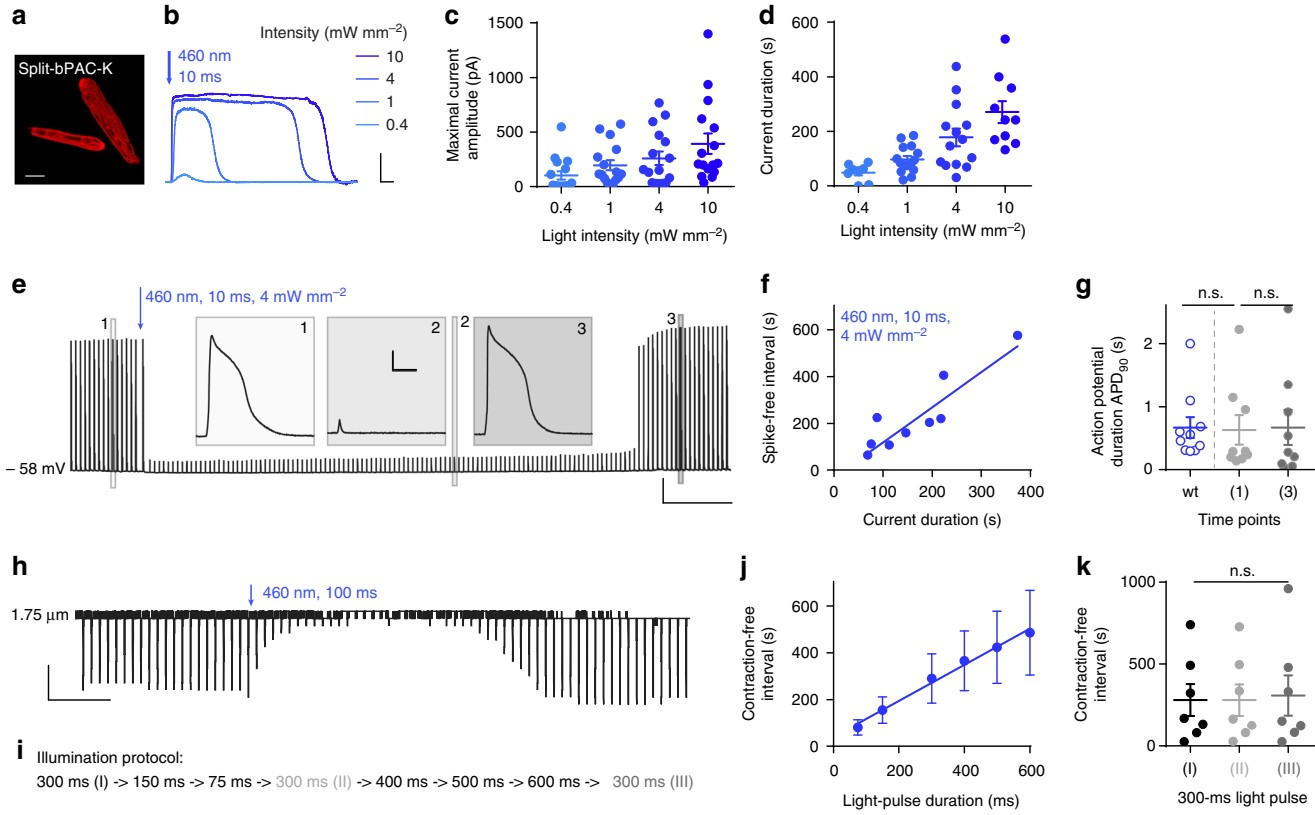

**Fig. 2** Split-bPAC-K efficiently silences primary ventricular cardiomyocytes. **a** Fluorescence of split-bPAC-K-mCherry expressed in cultured ventricular cardiomyocytes from rabbit. Scale bar: 20 μm. **b, c** Photocurrents and quantification of current amplitudes in isolated cardiomyocytes following blue light stimulation at different light intensities ($n = 16$ cells, seven cultures). Scale bars: 30 s, 100 pA. **d** Quantification of split-bPAC-K-mediated photocurrent duration (time from current onset to 50% current decline after maximum) at different intensities ($n > 11$ cells, 3–10 cultures). **e** Inhibition of electrically evoked APs (10 ms current ramp) by blue light pulse. Insets display response to electrical stimulation at three indicated time points. Scale bars: 30 s, 20 mV (overview trace); 100 ms, 20 mV (insets). **f** Duration of AP-free interval after split-bPAC-K activation by 10 ms light pulse as a function of corresponding photocurrent duration (460 nm, 4 mW mm$^{-2}$) ($n = 9$, six cultures). **g** Quantification of AP duration (APD$_{90}$) of wildtype (wt) and PAC-K expressing cardiomyocytes before and after light application ($n = 9$–10, four cultures). APD$_{90}$ before light (time point 1 in **d**) is not significantly different from APD$_{90}$ after inhibition (time point 3 in **d**) ($p = 0.3$, paired two-tailed $t$-test) and APD$_{90}$ of untransduced control cells ($p = 0.9$, unpaired two-tailed $t$-test) **h** Light-induced suppression of field stimulation-evoked cardiomyocyte contractions. Scale bars: 30 s, 0.05 μm. **i** Illumination protocol for consecutive inhibition of cardiomyocyte contractions. **j** Contraction-free intervals as a function of duration of activating light pulses ($n = 7$, three cultures). **k** The duration of the contraction-free interval induced by a 300-ms light pulse applied three times over the course of the test protocol (I–III in **i**) is not significantly different ($n = 7$, three cultures; $p$(II vs. I) = 1.0, $p$(III vs. I) = 0.5, one-way repeated measures ANOVA, Dunnett test). Error bars represent SEM

Importantly, split-bPAC-K was completely insensitive to the green light used for bReaChES activation. In contrast, blue light-triggered spiking via bReaChES activation, an effect that could be minimized by using UV light at low intensity for bPAC activation (Fig. 4d, e). In less invasive cell-attached recordings we further used independent optical excitation and inhibition by bReaChES and split-bPAC-K to demonstrate that endogenous ATP levels are sufficient for repetitive and stable silencing of bReaChES-driven AP firing (Supplementary Fig. 4).

To test neuronal silencing in a more complex environment, we injected AAVs encoding either split-bPAC-K (Fig. 5a–f) or fused-bPAC-K (Fig. 5g, h) into the hippocampus of wild type mice, or a Cre-dependent version of split-bPAC-K into the hippocampus of parvalbumin-Cre transgenic mice (Fig. 5a, c). Whole-cell recordings from both pyramidal and parvalbumin-positive interneurons in cornu ammonis area 1 (CA1) of acute hippocampal slices demonstrated that blue light triggers pro-longed hyperpolarizing photocurrents (Fig. 5b, d). Hyperpolarization was accompanied by a strong reduction in the input resistance, demonstrating that opening of SthK channels creates a shunting inhibition that will further reduce the excitatory drive (Fig. 5b, d). The photoactivated SthK current efficiently

suppressed ramp-induced neuronal firing in all neuron types tested (Fig. 5e, Supplementary Fig. 5). Of note, repetitive split-bPAC-K activation by 5 ms light pulses every 20 s persistently silenced AP firing for minutes (Fig. 5f).

Two-photon excitation laser-scanning microscopy in combination with fluorescent indicators allows high-resolution fluorescence imaging in intact tissues, and single-cell manipulation when applied for the activation of optogenetic actuators. It has therefore become a routine method for the interrogation of single-cell behavior in situ. In accordance with reported two-photon excitation cross sections of flavoproteins[41], we assessed whether we could activate fused-bPAC-K at 930 nm in individual CA1 pyramidal neurons in acute hippocampal slices. Neuronal firing was efficiently suppressed by a 50 ms-long two-photon scan on the somata of individual neurons (Fig. 5g, h).

**In vivo silencing of neuronal activity by bPAC-K.** Finally, we assessed the efficacy of the bPAC-K silencer system in vivo. Silicon probe recordings in anesthetized mice expressing split-bPAC-K in principal cells revealed reliable inhibition of CA1 unit activity by a single light flash (Fig. 6a–d). We also recorded neuronal activity by two-photon in vivo imaging via the

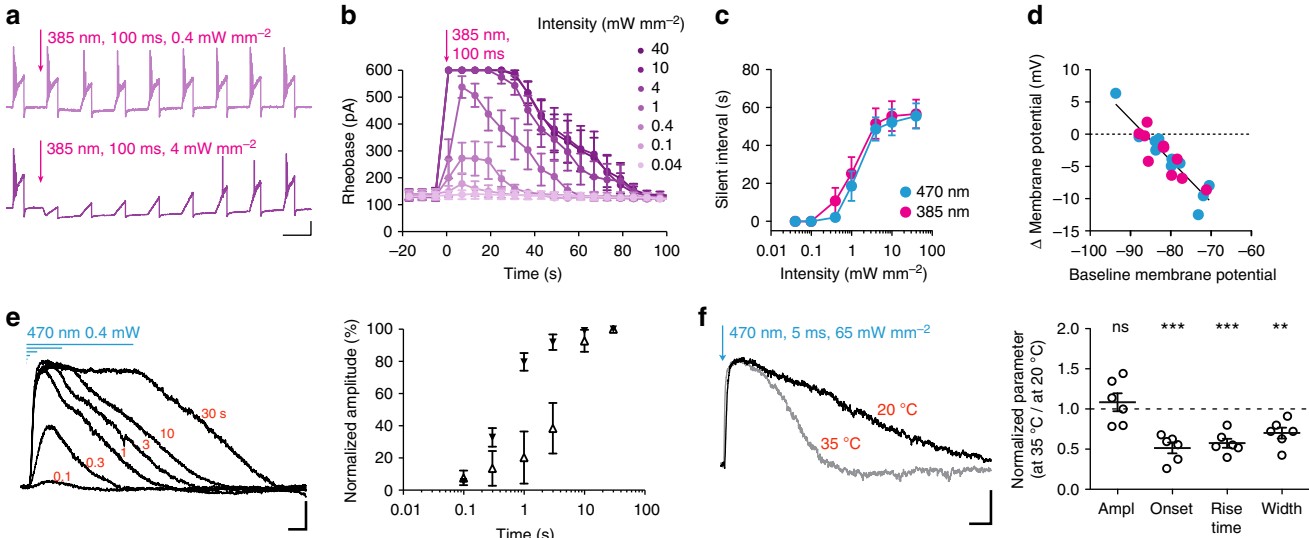

**Fig. 3** Split-bPAC-K-mediated silencing of neurons. **a** Example traces illustrating light intensity-dependent inhibition of current ramp-driven AP firing in a neuron expressing split-bPAC-K. Scale bars: 5 s, 20 mV. **b** Changes in the rheobase following split-bPAC-K activation ($n = 7$–8, three cultures; 600 pA reflects no spike). **c** The intensity of the bPAC activation light determined the duration of spike-free intervals ($n = 7$–8, three cultures). **d** Correlation of the extent of membrane potential changes induced by split-bPAC-K activation with the resting membrane potential before the light pulse (Pearson correlation, two-tailed: $p < 0.0001$, $R^2 = 0.83$). Data was pooled from experiments with 470 and 385 nm at 10 and 40 mW mm$^{-2}$. **e** Light pulse duration-dependent peak currents at light intensities of 0.4 mW mm$^{-2}$ (filled triangles) or 0.04 mW mm$^{-2}$ (open triangles; $n = 5$–7, two cultures). **f** Example traces of the SthK current recorded at near-physiological temperature (35 °C) or at room temperature (20 °C) from the same cell. The sequence of recordings at high and low temperatures was randomized between cells, and parameters were normalized to room-temperature conditions for each neuron ($n = 6$, three cultures). One sample $t$-test: $^{**}p < 0.01$; $^{***}p < 0.001$, ns: not significant. Scale bars in **e** and **f**: 5 s, 100 pA. Error bars in all graphs represent SEM

genetically encoded Ca$^{2+}$ indicator GCaMP6s[42] (Fig. 6e–h). GCaMP6s was excited at 1000 nm to avoid co-activation of bPAC[41]. Baseline activity of hippocampal neurons in mice expressing both fused-bPAC-K and GCaMP6s was not different compared to neurons lacking the actuator ($0.92 \pm 0.05$ vs. $1.06 \pm 0.1$ Hz), highlighting the absence of any identifiable dark activity. Wide-field illumination (488 nm) robustly and selectively reduced Ca$^{2+}$ transients in fused-bPAC-K-expressing neurons (Fig. 6f–h). In order to assess whether the PAC-K silencer is able to alter behavior in a well-characterized model animal, we targeted bPAC-K to zebrafish motor neurons using the UAS/Gal4 system (Fig. 7a–c), and studied spontaneously occurring muscle contractions (coils) of the embryos as a simple form of motor behavior. Under control conditions, embryos displayed coiling behavior at a frequency of 3–4 coils per 5 s. Blue light (6 μW mm$^{-2}$, 1 s) abolished coiling within 5 s after *light on*. About 20 s after *light off*, embryos started to move again and displayed transiently increased coiling behavior (ca. six coils per 5 s), compatible with a rebound from the inhibition state (Fig. 7d, g). Illuminating with further reduced intensity (2 μW mm$^{-2}$) for 15 s instead of 1 s did not significantly alter the effectiveness of inhibition ($n = 28$; $p > 0.05$; Fig. 7e, h). Non-expressing control fish showed a slight reduction of coiling activity after blue light application, which can be explained by photoinhibition of embryonic motor responses through endogenous, non-visual rhodopsins[43]. Consistently, both non-expressing and transgenic fish showed a similar response to a 625 nm light pulse (Fig. 7f, i). Taken together, bPAC-mediated activation of SthK mediates robust silencing of neurons in the vertebrate central nervous system.

## Discussion

Considerable efforts have been dedicated to developing tools for optogenetic inhibition of excitable cells[44]. Such tools are critical for understanding the roles of a particular cell type in the complex functional architecture of excitable tissues, such as the heart and the brain. For their effective use, optogenetic inhibitors must be (i) reliably targetable to the cells of interest, (ii) allow efficient and reversible inhibition of AP firing, ideally for variable durations, and (iii) have no dark activity or paradoxical effects on ion concentrations that may affect relevant cell and organ functions. The currently available inhibitory tools do not satisfy all of these conditions, in particular given that many experimental designs require prolonged inhibition. Different variants of chloride or proton pumps have been generated, with the general caveat that their prolonged activation may change intracellular ion concentrations, a phenomenon observed both for the proton pump Arch3[12] and the chloride pump eNpHR3.0[10]. This can lead to paradoxical effects, for instance through changes in the GABA reversal potential[11]. For both types of pumps, rebound depolarizations after termination of light exposition have been described[12]. Chloride-conducting channelrhodopsins (iChloc, iC++, and ACRs) have the advantage that, in contrast to ion pumps, they can produce inhibition by clamping the membrane potential to the chloride reversal potential[7–9]. However, activation of chloride channels suffers not only from the fact that prolonged chloride fluxes alter intracellular Cl$^-$ concentrations, but also from the observation that they can trigger depolarizing cation fluxes via voltage-dependent channels in neurons or neuronal compartments with a high intracellular Cl$^-$ concentration, based on depolarizing Cl$^-$ efflux upon light exposure[12,44,45]. Also the inhibition of myocardial excitability in this setting[46] is based on depolarization of cells above the excitation threshold, which is undesirable for rhythm management as it may lead to a potentially arrhythmogenic gain in cellular Ca$^{2+}$ uptake.

Optogenetic activation of K$^+$ conductances can potentially overcome these limitations, which has spurred the development of several light-gated K$^+$ channel systems[15–19]. The most promising published example is the engineered blue-light-induced

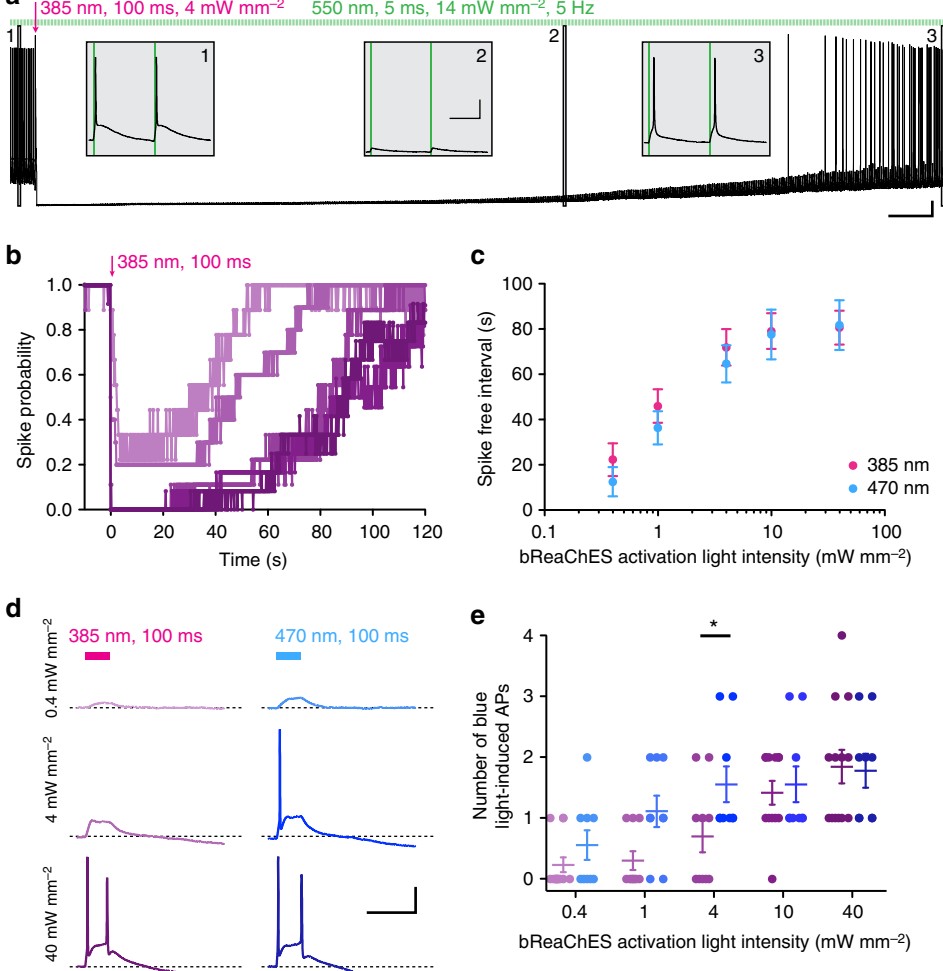

**Fig. 4** Bimodal optogenetic control of neurons expressing split-bPAC-K and red-shifted channelrhodopsins. **a** Dual-color optogenetic control with AP induction by bReaChES activation using 5 ms of 550 nm light at 5 Hz and split-bPAC-K-mediated hyperpolarization. Insets show currents preceding split-bPAC-K activation and currents at 18 and 95 s post-illumination. Scale bars: 5 s, 10 mV (overview trace); 100 ms, 20 mV (insets). **b** Intensity-dependent spike suppression of green light-triggered APs by 385 nm light ($n = 9$–12, five cultures; no error bars shown for clarity). **c** The duration of the spike-free interval in two-color excitation/inhibition experiments was independent of the wavelength used for split-bPAC-K activation (385 nm: $n = 9$–12, five cultures; 470 nm: $n = 9$, three cultures). **d** Intensity and wavelength-dependent cross activation of bReaChES by blue light. Dashed lines indicate resting membrane potential before the light pulse. Scale bars: 200 ms, 20 mV. **e** AP firing evoked by activation of bReaChES with 385 or 470 nm light was minimized when using low light intensities for bPAC activation (385 nm: $n = 10$–13, five cultures; 470 nm: $n = 9$, three cultures; Bonferroni's multiple comparison test, $^{*}p < 0.05$). Error bars represent SEM

K[+] channel (BLINK1) consisting of a small viral K[+] channel connected to a LOV-Jα photoreceptor module[14]. BLINK1 is not, however, robustly expressed in mammalian cells. In contrast, we demonstrate here that the SthK channel is well expressed in different cell types, and that the PAC-K constructs allow highly efficient hyperpolarization via a light-gated K[+] conductance in cell lines, cardiomyocytes, and neurons. Whereas ion pumps such as halorhodopsin or archaeorhodopsin require continued photon flux due to the 1:1 photon:ion coupling, PAC-K-mediated hyperpolarization can be maintained by repetitive brief light exposure at low frequencies, or continuous illumination at very low intensities. This allows cell re-polarization or hyperpolarization over prolonged periods with a minimal photon budget, avoiding tissue warming and altered blood flow[47–50]. One should note though that illumination intensity determines the current kinetics (Fig. 1, Supplementary Figs. 1 and 2), with higher intensity providing faster onset, but also longer duration of the hyperpolarization. Our tool is applicable on time scales from 100 ms up to minutes. However, the tool does not enable fast

manipulations in the millisecond range that might be required for rapid closed-loop optogenetic control of network activities at >10 Hz. Shorter effect durations can be achieved by using TpPAC instead of bPAC (Supplementary Fig. 3), but with similar onset kinetics at saturating light intensities (Fig. 1g). An additional advantage is the efficient activation of the PAC-K system by two-photon illumination. Since activation requires only milliseconds of illumination to trigger K[+] currents that last for seconds, PAC-K allows silencing of defined sets of individual neurons by sequential optical targeting. The PAC-K construct further provides the opportunity to combine optically controlled inhibition with excitation using red-shifted excitatory opsins, such as C1V1[51], Chrimson[52], ReaChR[53], or variants of these. Dual-color optogenetic excitation/inhibition experiments so far suffered from spectral overlap of the rhodopsin-based actuators, and required calibration of the light intensities for optimal bidirectional control of membrane voltage[54]. The specific activation of the PAC's BLUF domain with light of a wavelength shorter than 500 nm allows activation of red-shifted channelrhodopsins at maximum

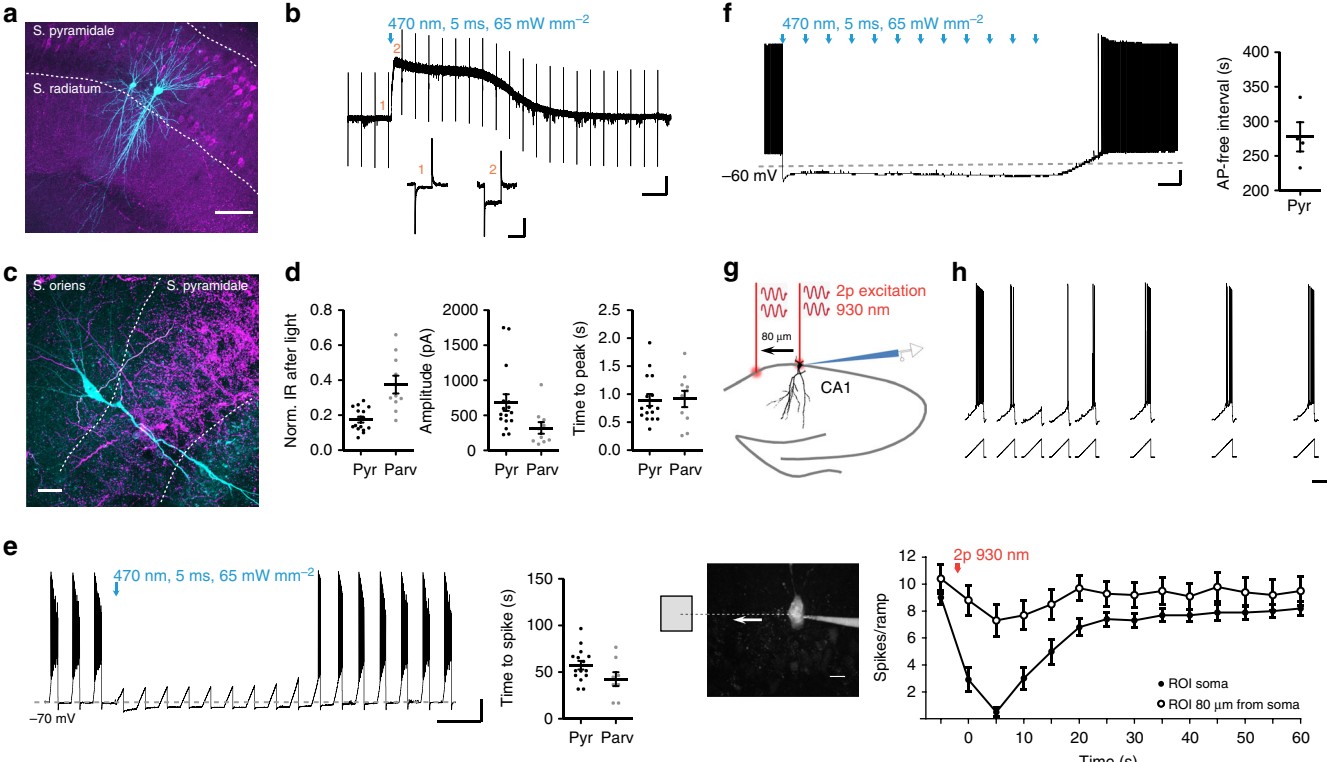

**Fig. 5** Silencing of neuronal activity in hippocampal slices. **a** Split-bPAC-K-mCherry-expressing pyramidal cells in area CA1. Recorded cells were filled with biocytin (green). Scale bar: 100 μm. **b** Blue light elicited outward current in a split-bPAC-K-positive pyramidal cell. Inset shows test pulse before (1) and after the light pulse (2). Scale bars: 10 s, 100 pA (overview trace); 50 ms, 100 pA (test pulses). **c** split-bPAC-K-mCherry expression in hippocampal slice of a parvalbumin-Cre transgenic mouse. Note the dense mCherry labeling in stratum pyramidale showing axons from parvalbumin-positive interneurons. Scale bar: 25 μm. **d** Quantification of the light-induced reduction in input resistance (IR), hyperpolarizing current amplitude, and time to peak after light pulse (pyramidal neurons: $n = 16$ from two mice; interneurons: $n = 10$ from three mice). **e** Suppression of current ramp-elicited APs by illumination. In pyramidal cells and parvalbumin-Cre$^+$ interneurons, first spikes reappeared after $58 \pm 5$ and $43 \pm 7$ s, respectively (pyramidal neurons: $n = 15$ from two mice; interneurons: $n = 8$ from three mice). Scale bars: 10 s, 20 mV. **f** Repetitive illumination (every 20 s for 5 ms) caused long-lasting but reversible inhibition of constant current-elicited spiking ($n = 4$, one mouse). Scale bars: 20 s, 10 mV. **g** Two-photon activation of fused-bPAC-K in CA1 neurons. The two-photon laser was either positioned over the soma of a fused-bPAC-K-positive neuron, or shifted ~80 μm to an adjacent area. Scale bar: 10 μm. **h** Inhibition of firing by the two-photon beam pointing at the soma (closed circles) or shifted from the soma (open circles, $n = 10$, three mice). Data in panel **a**-**f** was obtained with split-bPAC-K, while data in panel **g** and **h** was obtained with fused-bPAC-K. Scale bars: 1 s, 25 mV. Error bars in all graphs represent SEM

intensity. Still, the blue light used for PAC-K activation can potentially trigger an AP by activation of the red-shifted channelrhodopsin (Fig. 4e). However, given the extremely high light sensitivity of PAC-K (Fig. 1c) to blue and UV light, combined with the ability to integrate light over time (Fig. 3e), one can minimize activation of any co-expressed channelrhodopsin. Thus, bidirectional optical control of membrane voltage is feasible by expressing the actuator and inhibitor either in genetically distinct, but anatomically overlapping cell populations, or by co-expressing activator and inhibitor within the same cell type, as shown here for the combination of split-bPAC-K and bReaChES. Our new tool is useful in a variety of model systems and across disciplines exploring excitable cell systems, from the neurosciences to cardiovascular research.

An important aspect of a cAMP-based two-component optogenetic silencer is that one has to consider other cAMP-induced effects on intracellular signaling cascades, cellular metabolism, and gene expression. However, it should be noted that the PAC-K system was developed for silencing individual cells or cell populations in intact tissue, whether in neuronal networks or in the heart. These experiments allow one to study the functional circuitry constituted by the remaining non-affected cells, rather than the particular physiology of silenced cells. However, to exclude cAMP-mediated side effects in experiments involving multiple light applications over several days, it seems advisable to use PAC-only expression as an appropriate control for cAMP effects. bPAC-K represents a highly light-sensitive system due to the light-integration of bPAC and the intrinsic amplification by cAMP. Therefore, exposure of the PAC-K system to background light <500 nm should be minimized whenever possible. In cell types with low phosphodiesterase activity, baseline activity of the PAC might be of concern, although we found little dark activity across various experimental models, confirming previous reports[31,55]. If required, the bPAC(S27A) variant[39] with further reduced dark activity could be used, as the mutant enzyme is equally potent as wildtype bPAC in activating SthK (Supplementary Fig. 1). To conclude, the PAC-K silencer represents a novel genetically targetable light-controlled K$^+$ channel that is well expressed in a variety of model systems and cell types. It can thus be exploited for a wide range of scientific research questions for which efficient silencing of target cells is required.

## Methods

**Animal experiments**. All animal experiments were carried out according to the guidelines stated in Directive 2010/63/EU of the European Parliament on the protection of animals used for scientific purposes and were approved by the local

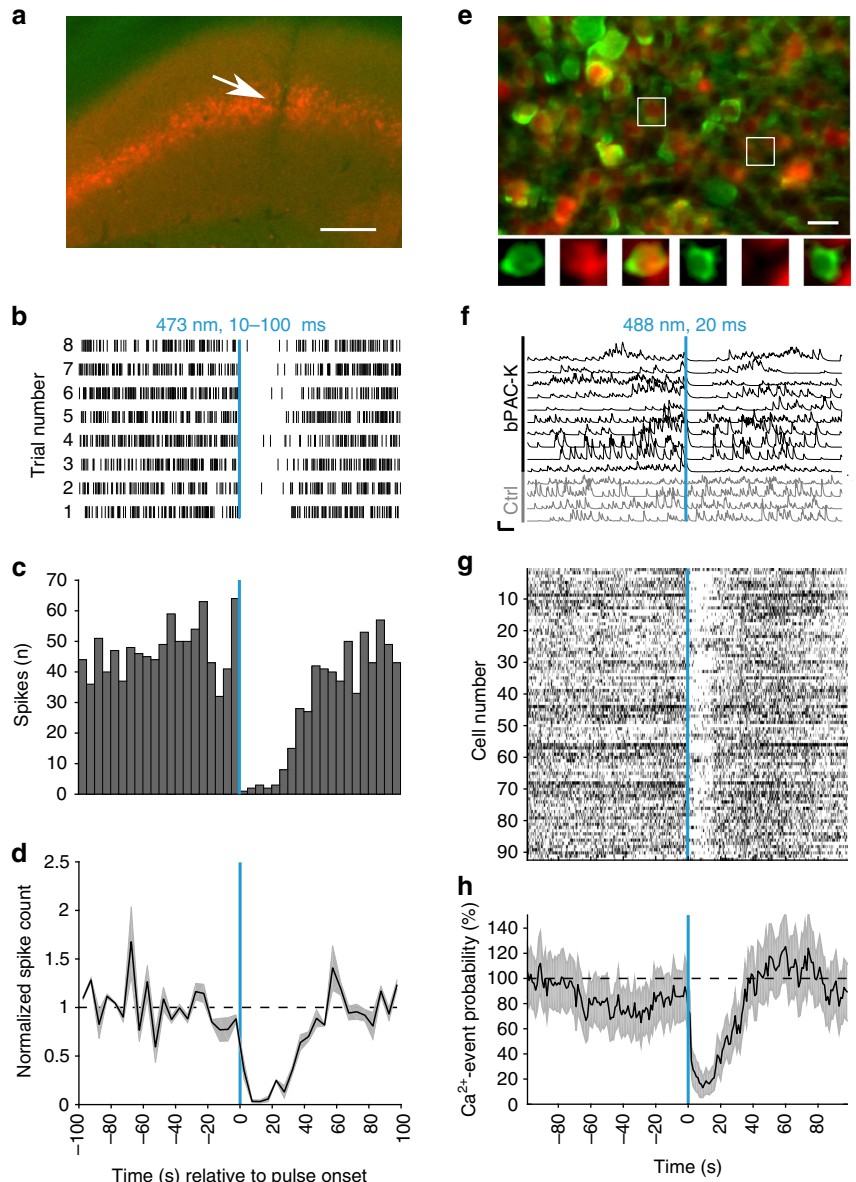

**Fig. 6** In vivo inhibition by the PAC-K silencer in mice. **a** Silicon probe track (arrow) in CA1 expressing split-bPAC-K/mCherry (red) in stratum pyramidale. Scale bar: 200 μm. **b** Multi-unit activity recorded under urethane anesthesia was strongly reduced following blue light application. Each trial represents a single light pulse (10 or 100 ms) and spikes (black lines) before and after this pulse. **c** Spike count histogram for spikes shown in **b**, summed over all trials (5 s time bins) from one recording. **d** Normalized spike counts averaged for three recorded mice. **e** In vivo two-photon image of CA1 pyramidal cells expressing GCaMP6s (green) and mCherry (red) indicating fused-bPAC-K-positive cells. Scale bar: 20 μm. Close-up views show a cell expressing both GCaMP6s and fused-bPAC-K/mCherry (left), and a cell expressing only GCaMP6s (right). **f** Representative Ca²⁺ fluorescence traces for individual pyramidal cells expressing fused-bPAC-K, and bPAC-K-negative cells (Ctrl). Scale bars: 10 s, 100% $\Delta F/F$. **g** Scatter plot of Ca²⁺ event onsets for 92 bPAC-K-expressing cells contained in a single field of view, overlaid for five successive trials. **h** Overall probability of Ca²⁺ events after light stimulation recorded from 274 fused-bPAC-K-positive cells ($n = 4$ mice). Error bars represent SEM

authorities in Baden-Württemberg (Regierungspräsidium Freiburg, X-16/10R), Bayern (Government of Upper Bavaria/Regierung Oberbayern, AZ.55.2-1-54-2532-101-12), Berlin (Berlin state government/Landesamt für Gesundheit und Soziales, G0092/15 and G150/17), and Nordrhein-Westfalen (Landesamt für Natur, Umwelt und Verbraucherschutz Westfalen (LANUV), AZ 84-02.04.2014.A254), Germany. No randomization or blinding was performed for the animals used in the study. We did not perform a power analysis to determine sample size prior to the experiments, since the aim of our study was to establish a new technology without prior knowledge on effect size and variability. Unless otherwise stated, all chemicals were purchased from Sigma Aldrich, Carl Roth, Tocris, and Thermo Fisher Scientific. Data are presented as mean ± SEM.

**Molecular biology**. SthK, bPAC, NgPAC1, TpPAC, OaPAC, and IcPAC (Acc. no.: WP051071283.1) coding DNA sequences were obtained by gene synthesis. Two construct configurations were used (Fig. 1a). In the cleavable configuration (split-PAC-K), the SthK channel is separated from the PAC by a 2A ribosome skip peptide from porcine teschovirus-1 (P2A)[56]. In the fusion protein variant (fused-PAC-K), SthK is linked to the C-terminus of the PAC (TpPAC or bPAC) by a peptide sequence (PRTYETSQVAPAGAP). In all cases, mCherry served as fluorescent marker. The wavelength used to excite mCherry did not overlap with the excitation spectrum of the PACs. Expression cassettes for PAC-SthK-T2A-mCherry and SthK-P2A-PAC-Cherry were subcloned into a pCS2-CMV-based plasmid for characterization in ND7/23 cells. Different PAC variants (Fig. 1b, c, Supplementary Fig. 1a, b) were compared using the cleavable PAC-K configuration.

For expression in cardiomyocytes, SthK-P2A-bPAC(wt)-mCherry or bPAC (S27A)-SthK-T2A-mCherry were cloned into pAdeno-CMV (Clontech) using the Adeno-X Adenoviral System (Clontech). Adenovirus was provided by the Viral Core Facility of the Charité—Universitätsmedizin Berlin, Germany.

Similarly, we designed two viral expression constructs for neuronal expression: For the split-PAC-K configuration, we inserted SthK-P2A-bPAC(wt)-mCherry or SthK-P2A-TpPAC-mCherry into the CW3SL AAV expression vector[57] containing

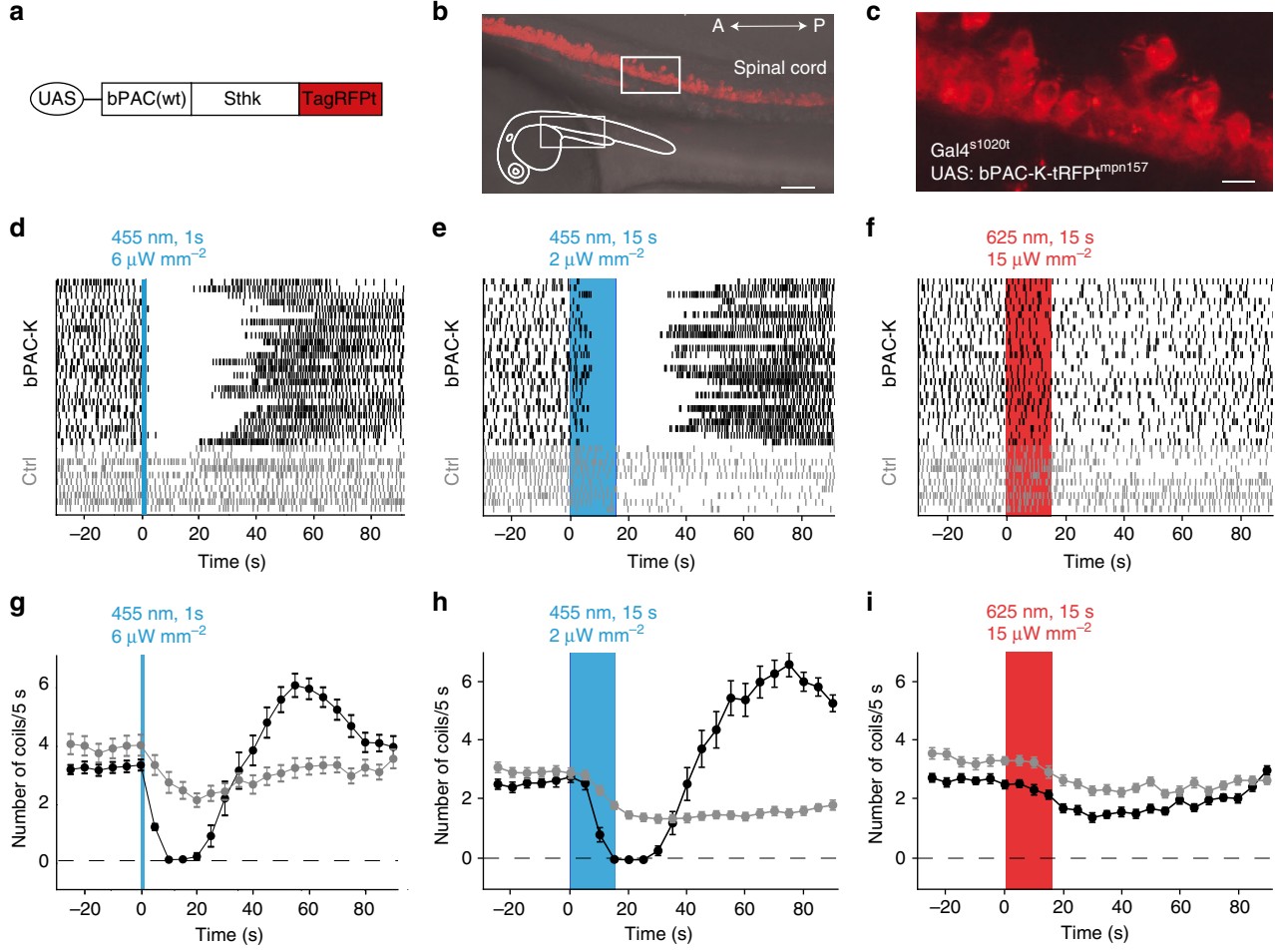

**Fig. 7** In vivo inhibition of coiling activity of zebrafish using fused-bPAC-K. **a** Schematic representation of the construct injected to generate the *UAS:bPAC-SthK-tRPFt^mpn157* transgenic line. The transgenic zebrafish expresses a fusion protein of bPAC(wt), SthK, and tagRFPt under the control of UAS. **b** Expression of *bPAC-SthK-tRFPt* in the spinal cord of a 24 h post-fertilization *Gal4^s1020t, UAS:bPAC-Sthk-tRFPt^mpn157* embryo. Scale bar: 50 μm. **c** High magnification of the spinal cord in fused-*bPAC-K* transgenic zebrafish. Scale bar: 10 μm. **d**–**f** Raster plots of coiling events in 25 fused-*bPAC-K* transgenic embryos (black) and 10 non-expressing embryos from the same clutch (gray) before, during, and after illumination with blue or red light. **g**–**i** Average number of coils per 5 s over three replicates in *fused-bPAC-SthK*-positive (black) and non-expressing (gray) embryos. All zebrafish embryos show low-level photoinhibition of motor responses due to endogenous light sensitivity. Error bars represent SEM

a small CaMKIIα promoter and a chimeric WPRE/poly-A element. For the fused-PAC-K configuration we created an AAV encoding bPAC(S27A)-SthK-T2A-mCherry, with a synapsin promoter driving expression of the construct. For experiments involving the red-shifted ChR bReaChES, EGFP-P2A-bReaChES was obtained by gene synthesis and inserted into CW3SL. SthK-P2A-bPAC-mCherry and EGFP-P2A-bReaChES were thus expressed by separate AAVs. For Cre-dependent expression of the PAC-K silencer, Sthk-P2A-bPAC-Cherry was inserted into a Flex-and-Excision (FLEX) cassette[58] downstream of a synapsin promoter. Adeno-associated viral particles of serotype 9 or 1 were produced according to published protocols[59]. Briefly, HEK 293 cells (ATCC CRL-1573) were transfected with a mix of pAD deltaF6, pAAV2/9, or pAAV 2/1, and the AAV expression vector. 48 h post transfection, cells were resuspended, collected by centrifugation, lysed in 50 mM Tris–Cl, 150 mM NaCl, and the viral particles were purified using an iodixanol gradient purification process. See Supplementary Table 2 for the use of the different constructs in each individual experiment.

**Combined imaging of cAMP levels and electrophysiology.** Viral transduction of the cAMP sensor was performed according to the manufacturer's protocol (Red Fluorescent cAMP assay, Montana Molecular). In brief, $9 \times 10^4$ HEK 293 cells were mixed with the BacMam stock, complete medium, and sodium butyrate and plated on coverslips in a multi-well plate. Cells were incubated for 30 min at room temperature in the dark and then for 24 h at 37 °C and 5% $CO_2$. After 24 h, cells were transfected with the bPAC-SthK fusion construct using Lipofectamine 2000 (Thermo Fisher), according to the manufacturer's protocol. 24–48 h after transfection, cells were transferred to a recording chamber placed on an inverted microscope (Olympus IX71). bPAC was stimulated using a Xenon discharge lamp (JML-C2,Rapp OptoElectronik) at 100 V passing through a 470 nm excitation filter

(Semrock, USA). Fluorescence of cADDis was recorded using a single-cell fluorometry system (Photon Technology International) with excitation light of 560 ± 3 nm; emission passed through a 641/75 nm filter (Semrock) onto a photon-counting photomultiplier tube. Currents were recorded in voltage-clamp mode (−40 mV) in the whole-cell configuration with an Axopatch 200B amplifier (Molecular Devices). The extracellular solution contained (in mM): NaCl 140, KCl 5.4, MgCl₂ 1, CaCl₂ 1.8, HEPES 5, glucose 10, adjusted to pH 7.4 with NaOH. The intracellular solution contained (in mM): NaCl 10, KAsp 130, EGTA 1, MgCl₂ 1, Na₂ATP 2, HEPES 10, adjusted to pH 7.2 with KOH.

**Electrophysiology in ND7/23 cells.** ND7/23 cells (ECACC 92090903, purchased from Sigma-Aldrich) were cultured in DMEM medium supplemented with 5% fetal calf serum (FCS). For electrophysiological recordings $5 \times 10^4$ cells were plated on 15-mm glass coverslips coated with Poly-D-Lysine. Cells were kept at 37 °C and 5% $CO_2$. After 24 h cells were transfected using Fugene® HD reagent at 3:1 reagent: DNA ratio according to the manufacturer's protocol (Promega).

Two days after transfection, whole-cell recordings in voltage-clamp mode were performed using a Multiclamp 700B amplifier (Molecular Devices). Signals were filtered at 4 kHz, digitized at 10 kHz, and recorded with Clampex 10.4. Electrodes with resistance between 2 and 3 MΩ were pulled in a horizontal micropipette puller P1000 (Sutter instrument) from borosilicate glass capillaries with filament (Harvard Apparatus; PG150T-15). An agar bridge containing 140 mM NaCl was used with the reference electrode. While recording, cells were maintained at room temperature and clamped at −40 mV. Extracellular buffer contained (in mM): 10 HEPES, 140 NaCl, 2.4 KCl, 2 CaCl₂, 4 MgCl₂, 10 glucose, pH 7.4, 300 mOsm. Intracellular buffer contained (in mM): 17.8 HEPES, 135 K⁺ gluconate, 4.6 MgCl₂, 4 MgATP, 0.3 NaGTP, 1 EGTA, 12 Na₂-phosphocreatine, 50 phosphocreatine

kinase, pH 7.3, 300 mOsm. pH was adjusted with N-Methyl-D-glucamine NMG. Light stimulation was provided by a Mightex light source coupled to an Olympus IX-73 microscope. PACs were illuminated for 10 or 100 ms with a 470 nm LED through a LUCPLFLN ×60/0.7 Olympus objective. Illumination field was 0.102 mm$^2$. Light intensities ranging from 0.11 to 100 mW mm$^{-2}$ were applied for light titration curves and the data was fitted using a logistic function. Maximum current was defined as the maximum current amplitude obtained when saturation of the channel was achieved. The slope of each current was calculated from a linear fit made from the signal onset (3× signal noise) to the half peak amplitude. Latency was defined as the time between the beginning of the light stimulus and the signal onset. Data was analyzed using Clampfit 10, Origin 9, and GraphPad Prism 5. Two-group comparison statistics were performed using the Wilcoxon Rank Sum Test.

Action spectra for bPAC, TpPAC, and bReaChES were measured on an Axiovert 100 Carl Zeiss microscope. Light was delivered through a W Plan-Apochromat ×40/1.0 DIC objective (Carl Zeiss) with a Polychrome V light source (TILL Photonics). Illumination field was 0.066 mm$^2$. The half bandwidth was set to ±7 nm for all data points, and light exposure time was controlled with a shutter system (VS25 and VCM-D1, Vincent Associates). In order to obtain the same photon irradiance for all wavelengths (Photon count: $2.34 \times 10^{13}$ photons s$^{-1}$), a motorized neutral density filter (NDF) wheel (Newport) was placed between the Polychrome V and the microscope. When necessary, a filter between the polychrome and the NDF was employed using a motorized 12-position filter wheel (Thorlabs).

**Cell isolation and culturing of primary cardiomyocytes.** Rabbits (9/10 weeks of both sex, total $N = 16$) were anesthetized via intramuscular injection of ketamine (12.5 mg/kg body weight)/xylazine 2% (0.2 ml/kg body weight). During anesthesia 1000 units heparin and 0.5 mg ketamine were given intravenously. Thiopental (12.5 mg/ml) was injected intravenously until apnea. The heart was excised, washed in base solution (in mM: 137 NaCl, 4 KCl, 10 HEPES, 10 creatine, 20 taurine, 10 glucose, 1 MgCl$_2$, 5 adenosine, 2 L-carnitine, 1 CaCl$_2$, 5 units/ml heparin-sodium, pH 7.4, 37 °C) including 1000 units heparin. The aorta was cannulated, the heart transferred to a Langendorff perfusion setup and it was perfused with base solution to wash out the blood. The heart was then perfused with calcium-free solution (in mM: 137 NaCl, 14 KCl, 10 HEPES, 10 creatine, 20 taurine, 10 glucose, 1 MgCl$_2$, 5 adenosine, 2 L-carnitine, 0.096 EGTA, pH 7.4, 37 °C) and finally digested with enzyme solution (calcium-free solution with 0.1 mM CaCl$_2$, collagenase type 2 (0.6 g/l, 315 U/mg) and protease XIV (0.03 g/l)). Left ventricle and septum were separated and the tissue was pulled apart in blocking solution (calcium-free solution with 0.1 mM CaCl$_2$, 0.5% bovine serum albumin).

After digestion cell suspensions were filtrated using a 1-mm mesh and centrifuged for 2 min at 22×g. Fibroblasts remaining in the supernatant were removed and cardiomyocytes were resuspended in plating medium (5 mM creatine, 2 mM L-carnitine hydrochloride, 5 mM taurine, 1 mM sodium pyruvat, 0.25 U/l insulin, 10 μM cytosine-β-D-arabinofuranoside, 5% FCS, and 0.05 mg/ml gentamycin). Cardiomyocytes were cultured on laminin-coated (100 μg/ml) coverslips. Medium was exchanged at 3–4 h after seeding cells, and adenovirus (type 5) coding for either of the two constructs added immediately after medium exchange (MOI75 for 48 h). Medium was renewed after 48 h and cells were recorded using the patch-clamp technique.

**Patch-clamp recordings and contraction tracking of cardiomyocytes.** Whole-cell patch-clamp measurements on single isolated cardiomyocytes were performed at room temperature using an inverted DMI 4000B microscope (Leica Microsystems), an Axopatch 200B amplifier, and an Axon Digidata 1550A (Molecular Devices). Activation light was delivered by a 460 nm LED using a 460–480 nm bandpass excitation filter and controlled via custom-built hardware (Essel Research and Development). External bath solution contained (in mM): 140 NaCl, 5.4 KCl, 1 CaCl$_2$, 2 MgCl$_2$, 10 glucose, 10 HEPES, pH 7.4, 300 mOsm; internal solution (in mM) 50 KCl, 80 K-aspartate, 2 MgCl$_2$, 3 Mg-ATP, 10 EGTA, 10 HEPES, pH 7.4, 300 mOsm. In current-clamp mode APs were triggered by current injections of 50% more than the threshold to elicit an AP, using a current ramp from 0 pA to 50% above threshold within 10 ms. Light pulses (460 nm/10 ms) were applied after 15 current injections and recordings were continued until APs reappeared. The AP duration (APD$_{90}$) was determined using the 90th percentile of the signal above resting potential as described[60]. Depolarizations with peak amplitudes below 0 mV were marked as inhibited. In voltage-clamp mode cardiomyocytes were clamped to −60 mV and light (460 nm/10 ms) was applied after 5 s.

Cardiomyocyte contractions were followed by detecting changes in sarcomere length. To this end, cardiomyocytes were imaged with a MyoCam-S camera (IonOptix) on an inverted microscope (DM IRBE, Leica Microsystems). Sarcomere length was calculated in real time by a Fast Fourier Transform of the power spectrum of the striation pattern (IonWizard, IonOptix)[61]. Cells were field stimulated at 14 V/0.25 Hz with a Myopacer (IonOptix). A blue light pulse (460 nm/100 ms) was applied after 88 s. Data was analyzed with pClamp 11, OriginPro, GraphPad Prism 7 and Matlab.

**Electrophysiology on neuronal cultures.** Hippocampal neurons prepared from P0 C57/BL6-N mice of either sex were grown on a glial cell feeder layer at a density of $1.25 \times 10^4$ cells cm$^{-2}$ in 24-well plates in Neurobasal-A supplemented with 2% B27 and 0.2% penicillin/streptomycin (Invitrogen). AAV particles were added 1–4 days after plating at a final concentration of $4.5 \times 10^9$ viral particles/ml for split-bPAC-K and $1 \times 10^9$ particles/ml for bReaChES. Cells were used for recordings after 15–21 days in vitro.

Whole-cell and cell-attached recordings were performed on an Olympus IX73 inverted microscope using a Multiclamp 700B amplifier (Molecular Devices) under the control of Clampex 10 (Molecular Devices). Data was acquired at 10 kHz and filtered at 3 kHz. Extracellular solution contained (in mM): 140 NaCl, 2.4 KCl, 10 HEPES, 10 glucose, 2 CaCl$_2$, and 4 MgCl$_2$ (pH adjusted to 7.3 with NaOH, 300 mOsm). Synaptic currents were blocked by 5 μM NBQX and 4 μM gabazine in most recordings. The intracellular solution contained (in mM): 135 K$^+$ gluconate, 17.8 HEPES, 1 EGTA, 4.6 MgCl$_2$, 4 Na$_2$-ATP, 12 disodium creatine phosphate, and 50 U/ml creatine phosphokinase, pH adjusted to 7.3 with KOH, 300 mOsm. Membrane potential was set to −60 mV in whole-cell voltage-clamp recordings. In current-clamp recordings, membrane potential was manually adjusted when cells depolarized to more than −55 mV. Data was not corrected for a liquid junction potential, unless for the analysis of the K$^+$ reversal potential in Fig. 3d (liquid junction potential of 22.6 mV). Recordings were terminated when holding currents were >−200 pA. Pipettes contained extracellular solution for cell-attached recordings. A TTL-controlled LED system (pE4000, CoolLED) was coupled into the back port of the IX73 microscope by a single liquid light guide. Fluorescence light was filtered using single band pass filters of a "Pinkel" quadband filter set (AHF F66-415) and delivered through an Olympus UPLSAPO ×20, 0.75 NA objective. Neutral density filters were used for experiments involving low-intensity light stimulations. Illumination intensities above the objective were measured prior to the experiments using a photodiode power sensor (Thorlabs S170C). bPAC was activated by flashes from the 385 or 470-nm LED, while bReaChES-triggered APs were elicited by 5-ms flashes of a GYR LED passed through a 554/24-nm filter.

Experiments involving temperature shifts were performed on an Olympus BX51 upright microscope equipped with a TC01 temperature controller (Multichannel Systems) and a CoolLED pE3 LED system. Excitation light of 470 nm was passed through a 470/40-nm filter (AHF F56-319) and applied via a 60× Olympus LumPlan FI 60 × 0.9 NA water immersion objective. Data was analyzed using AxographX 1.6.

**Recordings on hippocampal slices using bPAC-K.** Wild type and parvalbumin-Cre[62] mice (P26–P84) of either sex were injected in the hippocampal formation targeting area CA1 and the dentate gyrus with either AAV9.CamKIIα:SthK-P2A-bPAC-mCherry, AAV2.hSyn.bPAC(S27A)-SThK-T2A-mCherry, or AAV9.hSyn: FLEX(SthK-P2A-bPAC-mCherry), respectively. After 3–4 weeks, acute horizontal slices of the hippocampal formation were prepared. In brief, animals were anesthetized and decapitated. Under continuous safe light illumination, brains were quickly removed and placed in ice-cold artificial cerebrospinal fluid (ACSF) bubbled with carbogen (pH 7.4) containing (in mM): 87 NaCl, 26 NaHCO$_3$, 10 glucose, 2.5 KCl, 3.5 MgCl$_2$, 1.25 NaH$_2$PO$_4$, 0.5 CaCl$_2$, and 75 sucrose. Tissue blocks containing the hippocampal formation were mounted on a vibratome (Leica VT 1200S, Leica Microsystems), cut at 300-μm thickness, and incubated at 35 °C for 30 min. The slices were then transferred to carbogen-bubbled ACSF containing (in mM) 119 NaCl, 26 NaHCO$_3$, 10 glucose, 2.5 KCl, 2 CaCl$_2$, 2 MgSO$_4$, and 1.25 NaH$_2$PO$_4$. The slices were stored at room temperature in a submerged chamber for 1–5 h before being transferred to the recording chamber.

Whole-cell voltage and current-clamp recordings were performed from the hippocampal CA1 and CA3 area on an Olympus BX51 upright microscope. Recording electrodes with 4–5 MΩ resistance were pulled from borosilicate glass capillaries (Harvard Apparatus, Kent, UK; 1.5 mm OD) using a micropipette electrode puller (DMZ Universal Puller). The intracellular solution contained (in mM): 130 K-gluconate, 7 KCl, 10 HEPES, 4 Mg-ATP, 0.3 Na-GTP, 10 Na-phosphocreatine, and 0.2% (w/v) biocytin, pH 7.25 adjusted with KOH, 300 mOsm or K$^+$ gluconate 127, KCl 20, HEPES 10, EGTA 0.16, Mg-ATP 4, Na$_2$-ATP 2, D-glucose 10 (pH 7.25 (KOH), 295 mOsm). Membrane potential was set to −60 mV in whole-cell voltage-clamp recordings and at resting membrane potential in current-clamp recordings. Infected cells were identified by mCherry fluorescence. Excitation light of 470 nm from a CoolLED pE2 LED system was passed through a filter (AHF F56-319) and applied via a ×60 Olympus LumPlan FI 60 × 0.9 NA water immersion objective. Data was acquired using a Multiclamp 700B or BVC-700A amplifier (Molecular Devices or Dagan, respectively) under the control of Clampex 10.0 (Molecular Devices) at 10 kHz and filtered at 3 kHz. Recordings were analyzed using Clampfit (Molecular Devices) or AxographX 1.6.

A subset of electrophysiologically characterized neurons filled with 0.2% biocytin were fixed overnight in 4% paraformaldehyde dissolved in phosphate-buffered saline (PBS, pH 7.2). Recorded cells were visualized by 24 h incubation in PBS supplemented with streptavidin-coupled Alexa 488 (Invitrogen). After washing, the slices were mounted on slides and covered with Mowiol 4-88 (Carl Roth). Images were aquired using a Leica SP5 confocal laser scanning system.

Two-photon photostimulation was performed using a microscope equipped with a galvometer-based scanning system (Prairie Technologies, Ultima). A ROI was defined covering the neuronal soma and stimulation was carried out by

scanning the ROI with an ultrafast, Ti:sapphire pulsed laser (Chameleon Ultra, Coherent) tuned to 930 nm (50 ms scan duration, laser power <20 mW).

**In vivo Ca$^{2+}$ imaging**. We used 6–12-week-old wild type C57Bl/6 mice of both sex for all two-photon experiments. To express fused-bPAC-K and GCaMP6s in CA1 pyramidal cells, we co-injected both AAV2.hSyn.bPAC-SthK-T2A-mCherry. WPRE.SV40 and AAV1.Syn.GCaMP6s.WPRE.SV40[42]. Coordinates for stereotactic virus injection were (from Bregma in mm): −2.0 AP, −1.4 ML, and −1.4 DV.

Cranial window preparations followed a former description[63] and were performed under isoflurane anesthesia (2% in oxygen) as follows. Eyes were covered with eye-ointment (Bepanthen, Bayer) to prevent drying and body temperature was maintained at 37 °C using a regulated heating plate (TCAT-2LV, Physitemp) and a rectal thermal probe. The cranial bone was exposed, cleaned from connective tissue, and covered with optical adhesive dental cement (OptiBond FL, Kerr). A flat custom-made head post ring was fixed with UV curable dental acrylic (Tetric EvoFlow, Ivoclar Vivadent) to allow a close approach of the objective to the skull. A circular craniotomy (⌀ 3 mm) was opened above the right hemisphere hippocampus using a dental drill. Cortical tissue was aspirated using a 27-gauge needle until the fiber tracks of the corpus callosum became visible. A custom-made cone-shaped silicon inset (upper diameter 3 mm, lower diameter 2 mm, RTV 615, Movimentive) covered by a cover glass (⌀ 5 mm, thickness 0.17 mm) was inserted and fixed with dental acrylic. This special window design allowed easy implantation and maintenance, and the geometry was optimal for conserving the numerical aperture of the objective. The imaging session was conducted under anesthesia directly after surgery.

We used a commercially available two-photon microscope (A1 MP, Nikon) equipped with a 25× long working distance, water immersion objective (NA = 1, WD = 4 mm, XLPLN25XSVMP2, Olympus) controlled by the NIS-Elements software (Nikon). GCaMP6s was excited at 1000 nm using a Ti:sapphire laser system (~60 fs laser pulse width; Chameleon Vision-S, Coherent). At 1000 nm, flavoproteins are not excited by two-photon lasers[41], allowing separate excitation of the GCaMP with no cross-talk to the silencer. Emitted photons were collected using gated GaAsP photomultipliers (H11706-40, Hamamatsu) that electrically protected the optical sensor during light stimulation and quickly (~μs) recovered sensitivity afterwards. Stimulation light was delivered using a 488-nm diode laser (Sapphire 488 LP, Coherent) through the microscope galvanometric scanning unit in parallel to the imaging. Activity was acquired with a resonant scanning system at a frame rate of 15 Hz and duration of 10 min per movie. A 20-ms stimulation light pulse was delivered in the middle of each movie and five movies were recorded for every field of view. For excitation of mCherry we used a fiber laser system at 1070 nm (55 fs laser pulse width, Fidelity-2, Coherent) and recorded a reference image for every field of view.

To remove movement artifacts caused by breathing, recorded movies were registered using the steady scan Matlab toolkit[64]. Individual cell fluorescence traces were identified and Ca$^{2+}$ events were deconvolved using a constrained non-negative matrix factorization-based algorithm[65]. For later analysis, only components found in all five movies and showing a signal above threshold in the reference image were taken into account. We binarized Ca$^{2+}$ events by taking the onset of every event independent of event amplitude. Event probability was calculated by summing up the binarized events of all cells in all five movies within time bins of one second. We calculated the activity onset after stimulus for each individual cell by averaging the time to the first identified event after stimulus from all five movies.

**In vivo silicon probe recordings**. We injected 3× 200–300 nl of AAV9.CamKIIα: SthK-P2A-bPAC-mCherry into the left hemisphere of C57Bl6N mice (P26–P41, both sex) at coordinates (in mm relative to Bregma) AP −2.0, ML −1.6, DV −2.1, −1.7, −1.3. Four to six weeks after the stereotaxic injection, mice were anesthetized first with isoflurane (~2% in oxygen) and then by two i.p. injections (~10–15 min intervals) of 10% w/v urethane (total dose 1.8–2 g/kg) while lowering the isoflurane level under constant checking of breathing pattern and reflexes. Once a sufficient level of anesthesia was reached, mice were inserted into a stereotaxic frame and a craniotomy was performed at the same site as the original stereotaxic surgery. A 32-channel silicon probe (A1x32-poly2/3-10 mm-50-177-OA32LP, Neuronexus) with an integrated light guide (105 μm core, NA 0.22), attached to a reference wire with an Ag/AgCl pellet placed on the skull, was then inserted into the brain to a depth of 1550–2450 μm. Recordings were performed at depths of 1550–1800 μm using an RHD 2132 amplifier chip with an integrated 16-bit analog-to-digital converter and an RHD2000 USB interface board (Intan Technologies, Los Angeles, CA, USA). The light guide was coupled to an MBL-III-473 laser (100 mW, CNI, Changchun, China) producing pulses of 10 or 100-ms duration. Since no differences in response were observed, we pooled trials with both durations. Inter-pulse intervals were at least 3 min. At the end of each experiment, mice were killed with an overdose of urethane. We then performed cardiac perfusion with 0.1 M PBS followed by 4% paraformaldehyde (PFA) before removal of the brain and overnight fixation in PFA. Coronal sections (100 μm) were cut on a vibratome and imaged with a fluorescence microscope (Leica) to allow identification of mCherry-labeled cells, as well as to visualize the track left by the silicon probe. We confirmed that recordings from all mice were acquired in the hippocampal area CA1, and that the recording sites overlapped with the mCherry-labeled injection sites. As the aim of

this experiment was to obtain a qualitative impression of the construct's effectiveness in inhibiting neuronal activity in mouse brain in vivo, and because we observed a very clear effect with the first three mice, no further animals were used. A more formal evaluation and quantification is beyond the scope of this study.

**Zebrafish experiments**. Zebrafish were raised in Danieau's solution (17 mM NaCl, 2 mM KCl, 0.12 mM MgSO$_4$, 1.8 mM Ca(NO$_3$)$_2$, 1.5 mM HEPES) at 28 °C in the dark to avoid unwanted activation of bPAC-SthK. The following transgenic lines were used: *Et(−0.6hsp70l:Gal4-VP16)$^{s1020t}$*[66] and *UAS:bPAC-SthK-tagRFPT$^{mpn157}$* (see below), referred to as *Gal4$^{s1020t}$* and *UAS:bPAC-K-tRFPt$^{mpn157}$* in the text and figures, respectively.

**Generation of bPAC-SthK-tRFPt$^{mpn157}$ transgenic zebrafish**. Sequence encoding bPAC(wt)-SthK was fused to TagRFPT and placed downstream of the MCS of the upstream activating sequence (*UAS*) containing the Tol2 transposon system (*pTol2-14xUAS*, Baier laboratory). The assembly was performed with the In-Fusion® PCR cloning system (In-Fusion HD Cloning Kit, TaKaRa Clontech, Cat. #639649). The gene encoding the fluorescent protein TagRFPT was amplified from the following plasmid: *UAS:lyn-tagRFPT*[67]. The construct (25 ng/μl) was injected into eggs with the TL genetic background at the one-cell stage together with zebrafish codon–optimized Tol2 transposase mRNA (50–100 ng/μl) synthesized from *pCS-zT2TP*[68].

**Confocal imaging of zebrafish embryos**. F1 embryos (22–25 hpf; sex not determined) were dechorionated and mounted in 2.5% low melting agarose in Danieaus solution. Fish were imaged with a Zeiss LSM700 confocal microscope with a ×20 water immersion objective.

**Inhibition of spontaneous movement in the coiling assay**. Zebrafish embryos were placed in a 48-well plate well, inside a dark, temperature-controlled behavior box. After several minutes of adaptation, spontaneous coiling movements were recorded using an IR-sensitive camera (Mikrotron EoSens). Three recordings were carried out for each condition, with 2-min intervals in between. Activation light for bPAC was provided by a blue (455 nm) or red (625 nm) LED (Thorlabs) placed ~15 cm above the samples and controlled by a custom-written Python code using a LabJack U3. Light intensity was measured at the location of the sample with an Energy Meter Console (PM100D, Thorlabs).

Image analysis was performed with custom-made python scripts (available upon request) following a published protocol as follows[69]. Single circular regions of interests (ROIs) were manually drawn around the chorion of each embryo. To identify movements of the embryo, every frame was subtracted from the previous frame. For each ROI, single coiling movements were identified from a series of frames in which the pixel values changed between frames above a pre-defined noise threshold. Embryos that did not show any coiling movement during the whole recording were removed from the analysis. The average coiling frequency was then calculated across embryos of a single trial and across the three trials, for each condition.

**Data collection and analysis**. Statistical analysis was performed in MATLAB, OriginPro, or GraphPad Prism v5-v7. Data are presented as mean ± standard error of the mean (SEM), if not stated otherwise, and were tested for normality and equal variance. Animal numbers were chosen due to electrophysiological standards to distinguish an in the forehand unknown effect size from chance level. Sample sizes were not predetermined by statistical methods but are similar to those reported in comparable previous publications[3,7]. For light titration experiments, different light intensities were applied in a randomized fashion to the recorded cells. Data collection and analysis were not performed blind to the conditions of the experiments.

**Code availability**. Custom python scripts for image analysis are available upon request.

## Data availability

The data that support the findings in this study are available from the corresponding author upon reasonable request. DNA constructs are available via Addgene at https://www.addgene.org/118274/, https://www.addgene.org/118275/, https://www.addgene.org/118276/, https://www.addgene.org/118277/, https://www.addgene.org/118278/. Sequences have been uploaded to GenBank under accession codes MK027462, MK027463.

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

## Acknowledgements

The authors thank Susanne Rieckmann, Anke Schönherr, Tharsana Tharmalingam, Cinthia Buchmann, Irene Arnold-Ammer, and Enrico Kühn for excellent technical assistance, and Tilo Mathes for the IcPAC sequence information. This study was supported by the Max Planck Society (A.M.F., S.M., H.Ba.) and by grants from the DFG: SFB 958 (D.S.), Exc 257 (D.S.), SFB 1089 (H.Be.), SPP 1926 (B.R.R., S.O., R.A.K., F.S.W., P.H., A.M.F., H.Ba.), SPP 1665 (P.H., D.S., Y.A.B.S.), the BMBF: 01GQ1420B (D.S.), a BIH Delbrück Fellowship (D.S., P.B.), the ERC: CardioNECT (R.A.K., P.K.), Stiftung Charité (P.B.), and the German–French Collaborative Projects of ANR/DFG ebGLUNet (H.Be.). P.H. is a Hertie Senior Professor for Neuroscience and supported by the Hertie Foundation.

## Author contributions

Y.A.B.S., B.R.R., A.M.F., P.K., H.B., F.S.W., P.H., H.B., R.S. and D.S. designed the experiments. F.S.W., B.R.R., R.S., Y.A.B.S. and A.M.F. created the DNA constructs. Y.A.B.S., B.R.R., M.P., A.M.F., R.A.K., S.M., D.H., N.M., P.B., J.J.T., S.O., W.B., P.K., H.B., F.S.W. and R.S. performed the experiments and analyzed the data. B.R.R., Y.A.B.S., A.M.F., P.K., H.Ba., F.S.W., P.H., H.Be., R.S. and D.S. wrote the manuscript and designed the figures.

## Additional information

**Competing Interests:** The authors declare no competing interests.

