## [Peer Review File · Nature Communications]

Reviewers' Comments:

Reviewer #1:

Remarks to the Author:

In this study, Bernal-Sierra et al. report on a novel engineered potassium-conducting optogenetic tool which allows rapid and highly light-sensitive silencing of neurons and other excitable cells using blue light. The study utilizes an innovative approach, linking together a cAMP-synthesizing enzyme (PAC) with a cAMP-dependent potassium channel (SthK) to achieve a hybrid, light-triggered potassium conductance. The authors characterize several configurations of PAC-SthK, testing different PAC variants in a variety of cell types, and thoroughly characterize the function of the most successful variants in cell lines, primary neurons, cardiomyocytes, acute hippocampal slices and in-vivo recordings from mouse hippocampus. The spectral separation of PAC from red-shifted channelrhodopsins will allow bidirectional modulation of neural activity, which has been difficult to apply with microbial opsins due to their low light sensitivity and spectral overlap.

This is a much-needed tool in the inhibitory optogenetic toolbox, as existing tools each have their own caveats and limitations. For example, light-activated ion pumps lead to non-physiological concentrations of the transported ion. Chloride-conducting channelrhodopsins, although powerful, can be complicated to use in certain neuronal compartments and in certain types of neurons (for example, developing neurons and some unique types of neurons such as the hypothalamic oxytocin or vasopressin cells). The previously-published BLINK channel (Cosentino et al., Science 2015) seems to suffer from severe expression issues in mammalian systems and has not yet been applied in any rodent study so far.

The study is very complete, including a detailed description of the steps taken to engineer the new tools, as well as a thorough characterization of their spectral and temporal properties and their impact on excitability in a wide array of experimental systems. The degree of silencing and the minimal amounts of light it requires are remarkable. This manuscript, and the tools described in it, are of great potential interest to a broad scientific audience, extending beyond neuroscience due to the applicability of these tools for other excitable cell types. I am enthusiastic about this work, and about its suitability for the readership of Nature Communications. I just have a few minor points that I would like to see the authors respond to prior to publication.

Specific comments:

1. The main potential limitation to the approach seems to be the "off-target" effects of elevated cAMP levels on neurons, independent of the SthK channel activation. Although other inhibitory optogenetic tools surely have effects on neuronal physiology beyond silencing (pH changes with Arch, chloride gradient changes with NpHR), these effects should be characterized in detail to aid potential users. Did the authors record from neurons that only express bPAC during blue light stimulation? Does this in itself trigger any measurable changes in neuronal or synaptic physiology?
2. The authors describe the bPAC variant S27A as having a lower dark activity, and demonstrate very similar functionality of this variant compared with the wild-type PAC. This construct is then used in the zebrafish experiments and in the acute hippocampal slice work. Is there any reason not to choose this variant over the wild-type to reduce potential long-term changes in expressing neurons? This should be mentioned in the Discussion.
3. Fig. 5: This figure shows that bPAC/K can be activated with two-photon illumination. What is the two photon cross-section of the flavin chromophore? It has been argued that such chromophores have significant one-photon absorption in the near-infrared, which might lead to single-photon activation of bPAC/K throughout the illuminated depth and not just at the focal plane. The figure (5g) shows the spatial specificity at the XY plane, but it would also be useful to test whether similar specificity can be achieved along the Z axis.

4. Fig. 3e (right) is missing a legend. What do the filled vs. empty triangles represent?
5. p. 8 – “binary optical control” – should probably be “bidirectional optical control”.
6. The patch-clamp recordings described in Supp. Fig. 2a indicate no significant changes in the intrinsic properties of neurons expressing bPAC/K. However, many of the experiments in the manuscript use the P2A or T2A versions (bPAC-K) – these should be characterized for effects on intrinsic properties as well.
7. Supplementary Fig. 7: Please specify which variant was used in these recordings, since Fig. 5 seems to include experiments using both bPAC/K and bPAC-K.
8. Could the authors comment on the effects of activating PAC/K in axonal terminals? Does the strong conductance of SthK block synaptic release by shunting action potentials?

Reviewer #2:

Remarks to the Author:

In this manuscript, Bernal-Sierra and coworkers engineer a genetically-encoded two-protein system to silence neurons in mammalian cells and animals with blue light. Their design is novel - it combines a previously characterized blue light-activated adenylyl cyclase (PAC) from the bacterium *Beggiatoa*, and the cAMP-activated K⁺ transporter SthK from the bacterium *Spirochaeta thermophila*. They construct a version of the system where the two genes are expressed as a fusion peptide and post-translationally cleaved, and a second version where they remain fused. The former appears to generally have superior performance. They perform a rigorous, if not fairly standard, set of dynamical and spectral characterizations of the system in cultured cells, cardiomyocytes, and neurons, and in vivo in zebrafish, and mouse neurons. Overall, the system appears to perform well in all of these contexts. They also combine the system with a red light activated channelrhodopsin. Despite that blue light cross-talks with the channelrhodopsin, they show that substitution of UV light overcomes this. In addition to these performance features, their system appears to overcome several major limitations of other blue light neuronal silencing systems, and thus would appear to be of significant interest to the neuroscience community. A few important questions notwithstanding, the paper definitely seems appropriate for publication in Nature Communications.

One note I have is that the paper is written almost exclusively for neuroscientists. I suppose that is fine, since it goes without saying that neural optogenetics is a very important field. However, if the authors were to add introductions of some of the concepts in the paper and why they are important (e.g. zebrafish coiling, two-photon), and jargon (e.g. "2P") it would make the paper more readable by a broader audience. Non-neuroscientists generally won't have much luck understanding the experimental designs nor interpreting the data, but I suspect some outside neuroscience would find this paper interesting nonetheless.

Major comment:

1. "At 50% light saturation, currents reached a maximum within approximately 12 seconds (Supplementary Data 1a)". 12s seems slow relative to the millisecond timescales on which neurons change their membrane potential. It's obviously fast enough for the cell culture experiments in the paper, to silence neurons in a mouse, and to inhibit a behavior (coiling activity) in zebrafish. But does this timescale impose limitations for some optogenetic applications? The authors should discuss the limitations of their system explicitly. Examples of important questions in neuroscience that could not be answered with these slow kinetics (if any exist) should be stated. Potential for improving these kinetics in future work should also be discussed.

2. The authors attempt to improve dark activity of their system in the first section, but it fails. However, they don't describe that dark activity is a problem. Is it a problem? Does it create limitations? This issue needs to be discussed explicitly.

Minor comments:

1. Zero rationale is given for why a split versus a fused system were tried. Rationale needs to be explained.

2. I found the nomenclature "PAC/K" for the split system and "PAC-K" for the fused system to be exceedingly subtle. It makes reading the paper difficult. I recommend the authors consider more distinct nomenclature.

3. What the authors mean by "...a 10 times lower dose-response relation" is unclear.

4. It is not clear what was actually done when they say "The small size of the PACKS allowed expression by a single AAV coding for different PACK variants". I interpreted this as they were delivering the split and fused versions, but reading on, it appears that's not true. This needs to be clarified.

5. It's not clear which PACK silencer they are referring to when they say "To test neuronal silencing in a more complex environment, we injected AAVs encoding a PACK silencer into the hippocampus of wild type mice..."

Reviewer #1:

This is a much-needed tool in the inhibitory optogenetic toolbox, as existing tools each have their own caveats and limitations. For example, light-activated ion pumps lead to non-physiological concentrations of the transported ion. Chloride-conducting channelrhodopsins, although powerful, can be complicated to use in certain neuronal compartments and in certain types of neurons (for example, developing neurons and some unique types of neurons such as the hypothalamic oxytocin or vasopressin cells). The previously-published BLINK channel (Cosentino et al., Science 2015) seems to suffer from severe expression issues in mammalian systems and has not yet been applied in any rodent study so far. The study is very complete, including a detailed description of the steps taken to engineer the new tools, as well as a thorough characterization of their spectral and temporal properties and their impact on excitability in a wide array of experimental systems. The degree of silencing and the minimal amounts of light it requires are remarkable. This manuscript, and the tools described in it, are of great potential interest to a broad scientific audience, extending beyond neuroscience due to the applicability of these tools for other excitable cell types. I am enthusiastic about this work, and about its suitability for the readership of Nature Communications. I just have a few minor points that I would like to see the authors respond to prior to publication.

– Thank you!

1. The main potential limitation to the approach seems to be the “off-target” effects of elevated cAMP levels on neurons, independent of the SthK channel activation. Although other inhibitory optogenetic tools surely have effects on neuronal physiology beyond silencing (pH changes with Arch, chloride gradient changes with NpHR), these effects should be characterized in detail to aid potential users. Did the authors record from neurons that only express bPAC during blue light stimulation? Does this in itself trigger any measurable changes in neuronal or synaptic physiology?

- We agree with the reviewer that the issue of cAMP is an important point to consider. We have therefore performed new experiments with expression of bPAC-only in dissociated neuronal cultures. This data is now shown in panel g and h of the Supplementary Fig. 2. In summary, we could detect only very small depolarizing currents in these cells after blue light exposure (<-40 pA, compared to +800pA when SthK is co-expressed), which are at least partially mediated by hyperpolarization-activated cyclic nucleotide-gated (HCN)-channels.
- We agree that long lasting elevation of cAMP will affect cAMP-related gene expression but as we stated in the discussion on page 9, this is of low relevance for cells that should be taken out of the cellular network. However, for a cell that should be re-analyzed after hyperpolarization, potential side effects surely should be taken into account, and we recommend expression of PAC-only as an appropriate control in the Discussion section on page 9.

2. The authors describe the bPAC variant S27A as having a lower dark activity, and demonstrate very similar functionality of this variant compared with the wild-type PAC. This construct is then used in the zebrafish experiments and in the acute hippocampal slice work. Is there any reason not to choose this variant over the wild-type to reduce potential long-term changes in expressing neurons? This should be mentioned in the Discussion .

- The lower dark activity of the bPAC(S27A) mutant was described in prior work from the Hegemann lab (Stierl et al., 2014, Biochemistry). Therefore, we have chosen this variant for the fusion construct in order to minimize background activation of the SthK. However, we

have not found any evidence for a superior performance of the construct incorporating bPAC(S27A) compared to the bPAC(wt). We have also not observed any significant differences between the kinetics of the PAC-K with bPAC(S27A) and bPAC(wt), as shown in the Supplementary Fig. 1c-f, neither any adverse or negative effects of the expression of either bPAC variant. However, in cells with low PDE activity bPAC(S27A) should be the first choice, as described now in the discussion on page 9.

- Of note, in the methods section of the initial submission it was stated incorrectly that the bPAC(S27A) variant was used in zebrafish. We are sorry for causing this mistake: we also used the bPAC(wt)-construct for the zebrafish experiments, and have corrected the text accordingly.

3. Fig. 5: This figure shows that bPAC/K can be activated with two-photon illumination. What is the two photon cross-section of the flavin chromophore? It has been argued that such chromophores have significant one-photon absorption in the near-infrared, which might lead to single-photon activation of bPAC/K throughout the illuminated depth and not just at the focal plane. The figure (5g) shows the spatial specificity at the XY plane, but it would also be useful to test whether similar specificity can be achieved along the Z axis.

- An analysis of the 2P crosssection of flavin adenine dinucleotide (FAD) with a maximum near 900 nm shows a strong red-shift compared to 1P absorption spectrum peaking at 475 nm (see below Fig. 1 in Huang et al. 2002, Biophysical Journal). As 2P activation was done in our neuronal experiments with 930 nm, this is far removed from the excitation wavelengths for 1P absorption. Thus, we consider a single-photon activation of the flavin chromophore unlikely. Of note, we did not observe any activation of the PACK silencer by the infrared illumination used for electrophysiology experiments in acute slice experiments.
- We agree that it would be nice to know if the z-axis specificity is similar, however, this is much more difficult to demonstrate as the excitation intensity decreases with increasing depth into the tissue. Rigorous examination of this question would therefore require adapting laser intensity to the differential attenuation. We suggest that to demonstrate spatial specificity, therefore, our approach to move the excitation ROI is the most straightforward.

Figure 1 Comparison of the 2P-excitation action cross section (σ_{2p} ; $1 \text{ GM}=10^{-50} \text{ cm}^4 \text{ s}$) spectra of NADH (green triangles) and NADPH (blue inverted triangles) to those of FAD (black squares) and LipDH (red circles; right axis). FAD and LipDH have additional 2P-excitation peaks around 900nm. The 1P-absorption spectra of NADH (blue line), FAD (black line), and LipDH (red line), arbitrarily scaled at twice the excitation wavelengths, are red-shifted related to their 2P counterparts. Error bars are standard deviations of σ_{2p} values determined using 8–10 excitation intensities at each wavelength. Samples are $0.9 \mu\text{M}$ fluorescein in water (pH 11); $59 \mu\text{M}$ LipDH in 0.1M potassium phosphate (pH 7.6) and 0.2mM EDTA; and $94 \mu\text{M}$ FAD, $563 \mu\text{M}$ NADH, and $448 \mu\text{M}$ NAD(P)H in Tris buffer (pH 7.6). Reference: <https://www.cell.com/biophysj/fulltext/S0006-3495%2802%2975621-X>

4. Fig. 3e (right) is missing a legend. What do the filled vs. empty triangles represent?

- Thanks for noting this. We have updated the figure legend accordingly.

5. p. 8 – “binary optical control” – should probably be “bidirectional optical control”.

- Done. Thanks.

6. *The patch-clamp recordings described in Supp. Fig. 2a indicate no significant changes in the intrinsic properties of neurons expressing bPAC/K. However, many of the experiments in the manuscript use the P2A or T2A versions (bPAC-K) – these should be characterized for effects on intrinsic properties as well.*

- We agree with the reviewer that this important control was missing. We have now added a new analysis of two cultures of neurons expressing either split-bPAC-K or fused-bPAC-K, the main constructs used in this work. For both constructs we see no change in the intrinsic parameters when compared to uninfected neurons (see Supplementary Fig. 2a).

7. *Supplementary Fig. 7: Please specify which variant was used in these recordings, since Fig. 5 seems to include experiments using both bPAC/K and bPAC-K.*

- We assume that the reviewer is referring to Supplementary Fig. 5. For the experiments in this figure, only bPAC-K (new nomenclature now: “fused-bPAC-K”) was used. We have updated the figure legend accordingly - thanks for pointing this out.

8. *Could the authors comment on the effects of activating PAC/K in axonal terminals? Does the strong conductance of SthK block synaptic release by shunting action potentials?*

- This is a very important point, as optogenetic axonal silencing is a much-needed tool for the neurosciences. In order to test for optical silencing of transmitter release, we established an all-optical approach for the stimulation and silencing of transmitter release from long-range projections (see figure below). We injected a mix of two AAVs unilaterally into the right hippocampus of P21-P28 mice. The first virus encoded the red-shifted Channelrhodopsin ChrimsonR together with tdTomato, while the 2nd virus encoded the split- or fused-bPAC-K construct. After 4-6 weeks following expression of the constructs, we performed electrophysiological experiments on the contralateral side of the hippocampus in area CA3/CA1, and used 5 ms-pulses of 550 nm light for stimulating transmitter release from the terminals. Unfortunately, our result did not yield any positive evidence for successful silencing of transmitter release by PAC-K. While the green light stimulation elicited reliable transmitter release from the commissural projections, blue light did not suppress release. So far, we have no evidence that the SthK-channel is trafficked at sufficient levels into the axons. Much more work is needed to be conclusive on this point, and we think that the development and testing of PAC-K for presynaptic inhibition is beyond the scope of our current study.

No evidence for silencing of synaptic transmission by PAC-K at commissural fibers in the hippocampus

a Illustration of the virus injection (1:1 mix of two AAVs encoding the PAC-K silencer and ChrimsonR-tdTomato) into the right hemisphere of 24 – 26 days old mice. **b** 3 to 4 weeks later, 300 μm thick coronal slices were prepared. While the right hemisphere showed very intense red fluorescence of the mCherry from the PAC-K and tdTomato from the ChrimsonR construct, a clear tract of commissural fibers was visible in the non-infected, left hippocampus. The recording electrode was placed in CA3 of the left hippocampus, and synaptic transmission was evoked by stimulating axons expressing ChrimsonR using 5 ms long flashes of 585 nm light. **c** Examples of light-evoked field excitatory postsynaptic potentials (fEPSCs) in CA3 before and after stimulation with blue light. Traces are averages of 3 stimulations before (black) and after (blue) the 470 nm light flash. **d** Blue light exposure intended to activate PAC did not cause any inhibition of synaptic transmission ($n = 3$ slices of 3 mice).

Reviewer #2:

Overall, the system appears to perform well in all of these contexts. They also combine the system with a red light activated channelrhodopsin. Despite that blue light cross-talks with the channelrhodopsin, they show that substitution of UV light overcomes this. In addition to these performance features, their system appears to overcome several major limitations of other blue light neuronal silencing systems, and thus would appear to be of significant interest to the neuroscience community. A few important questions notwithstanding, the paper definitely seems appropriate for publication in Nature Communications.

One note I have is that the paper is written almost exclusively for neuroscientists. I suppose that is fine, since it goes without saying that neural optogenetics is a very important field. However, if the authors were to add introductions of some of the concepts in the paper and why they are important (e.g. zebrafish coiling, two-photon), and jargon (e.g. "2P") it would make the paper more readable

by a broader audience. Non-neuroscientists generally won't have much luck understanding the experimental designs nor interpreting the data, but I suspect some outside neuroscience would find this paper interesting nonetheless.

- Thanks for the positive comments! We agree with the reviewer that we can do better in making the paper more accessible for a broader readership and have modified our manuscript accordingly. For example, we now emphasize the value of two-photon microscopy for single-cell interrogations in complex tissue on page 7, stating: “Two-photon excitation laser-scanning microscopy in combination with fluorescent indicators allows high-resolution fluorescence imaging in intact tissues, and single cell manipulation when applied for the activation of optogenetic actuators. It has therefore become a routine method for the interrogation of single-cell behavior *in situ*.”

1. "At 50% light saturation, currents reached a maximum within approximately 12 seconds (Supplementary Data 1a)". 12s seems slow relative to the millisecond timescales on which neurons change their membrane potential. It's obviously fast enough for the cell culture experiments in the paper, to silence neurons in a mouse, and to inhibit a behavior (coiling activity) in zebrafish. But does this timescale impose limitations for some optogenetic applications? The authors should discuss the limitations of their system explicitly. Examples of important questions in neuroscience that could not be answered with these slow kinetics (if any exist) should be stated. Potential for improving these kinetics in future work should also be discussed.

- Well taken! It is correct; at 50% light saturation the maximum current is reached at approximately 12 s. However at light intensities $>1.3 \text{ mW}\cdot\text{mm}^{-2}$ the latency of the current decreases considerably and the slope increases in a way that at least 90% of maximum current can be achieved in 3 s or less (Supplementary Fig. 1a and 2c).
- Importantly, we see that action potential spiking is immediately suppressed when bPAC-K is activated with intense light, both when APs were evoked by bReaChES activation (Figure 4b, spike interval 200 ms), and in spontaneously firing neurons (Figure 5f: spontaneous spiking frequencies before the light pulse were between 2.6 and 4.5 Hz, no spikes were detected after the blue light).
- We have also tested other PACs - e.g. TpPACK (see Figs. 1 and 2) - to implement different kinetics to our system. Using TpPACK considerably reduced the effect duration of the silencer, but not the onset-kinetics.
- We discuss this issue on page 9 now in more detail, stating: “One should note though that illumination intensity determines the current kinetics (Fig. 1, Supplementary Fig. 1 and 2), with higher intensity providing faster onset, but also longer duration of the hyperpolarization. Our tool is applicable on time scales from 100 ms up to minutes. However, the tool does not enable fast manipulations in the millisecond range that might be required for rapid closed-loop optogenetic control of network activities at $>10 \text{ Hz}$. Shorter effect durations can be achieved by using TpPAC instead of bPAC (Supplementary Fig. 3), but with similar onset kinetics at saturating light intensities (Fig. 1g).”

2. The authors attempt to improve dark activity of their system in the first section, but it fails. However, they don't describe that dark activity is a problem. Is it a problem? Does it create limitations? This issue needs to be discussed explicitly.

- In our hands, dark activity was not a major problem, but this will depend on the model system and the experimental conditions used. We now discuss the issue of

background activity of the PAC (which could be due to dark activity or unintended activation by stray light) in detail on page 9 of our discussion-section: “bPAC-K represents a highly light-sensitive system due to the light-integration of bPAC and the intrinsic amplification by cAMP. Therefore, exposure of the PAC-K system to background light <500 nm should be minimized whenever possible. In cell types with low phosphodiesterase activity, baseline activity of the PAC might be of concern, although we found little dark activity across various experimental models, confirming previous reports^{29,53}. If required, the bPAC(S27A) variant³⁷ with further reduced dark activity with further reduced dark activity could be used, as the mutant enzyme is equally potent as wildtype bPAC in activating SthK (Supplementary Fig. 1).”

Minor comments:

1. Zero rationale is given for why a split versus a fused system were tried. Rationale needs to be explained.

- The fused system was intended to solve possible problems with cytoplasmic cAMP diffusion because the cyclase would be localized at the membrane close to the SthK channel. However, there were no significant differences in latency between fused vs. split construct as observed in Fig. 1g.

2. I found the nomenclature "PAC/K" for the split system and "PAC-K" for the fused system to be exceedingly subtle. It makes reading the paper difficult. I recommend the authors consider more distinct nomenclature.

- Thanks for pointing this out! We initially coined the very similar termini because for most experimental systems the two constructs can be used interchangeably. However, we agree with the reviewers point, and have changed the nomenclature now to split-PAC-K for PAC/K, and fused-PAC-K for PAC-K.

3. What the authors mean by "...a 10 times lower dose-response relation" is unclear.

- TpPAC-K had a shift of the dose-response curve to light intensities 10 times higher than the light intensities needed to activate PAC-K. We therefore now state on page 4: “TpPAC displayed 4 times faster off-kinetics and a 10 times lower light sensitivity.”

4. It is not clear what was actually done when they say "The small size of the PACKS allowed expression by a single AAV coding for rate different PACK variants". I interpreted this as they were delivering the split and fused versions, but reading on, it appears that's not true. This needs to be clarified.

- The sentence was intended to describe the advantage of the small cDNA size encoding both bPAC and SthK, which allows the two components to be expressed from a single AAV construct. This is true for both the split and the fused-PAC-K system. We have updated the sentence now, stating: “The small size of the PAC-K-encoding DNA sequences allowed gene delivery by a single adeno-associated virus (AAV).”

5. *It's not clear which PAK silencer they are referring to when they say "To test neuronal silencing in a more complex environment, we injected AAVs encoding a PAK silencer into the hippocampus of wild type mice..."*

- This statement introduces the experiments performed in Figure 5 (Silencing of neuronal activity in hippocampal slices), for which we used both the split- and the fused-PAK-K construct. Both constructs showed very similar effects (see also Supplementary Fig. 5). We now specifically state which construct was used for the experiments, both in the figure legend and also in the text. (p6 :“To test neuronal silencing in a more complex environment, we injected AAVs encoding either split- (Fig. 5a-f) or fused- (Fig. 5g,h) bPAK-K into the hippocampus of wild type mice, or a Cre-dependent version of split-bPAK-K into the hippocampus of parvalbumin-Cre transgenic mice (Fig. 5a, c).”)

Reviewers' Comments:

Reviewer #1:

Remarks to the Author:

The revised manuscript by Bernal et al. is much improved compared with the original submission. The authors have addressed all of the points raised in the review, and now provide sufficient experimental detail for this approach to be replicated and applied in various experimental systems. The standardization of the nomenclature used to describe the PAK constructs is appropriate.

The authors point out correctly that a PAC-only construct can be used as the "optimal" experimental control for PAC/K-expressing animals. I suggest that such a construct be made available (on Addgene or directly from the authors) so that it can be used by labs wishing to make use of the PAC/K technology.

Minor point: in page 9, the sentence "with further reduced dark activity" appears twice.

Reviewer #2:

Remarks to the Author:

The authors have done a nice job addressing my comments - except for the rationale behind the design of the split and fused PAC-SthK systems, which they only addressed in the reviewer response document. They should add a rationale around lines 108-110 of the main text. Pending this addition, I support publication of the manuscript in Nature Communications.

REVIEWERS' COMMENTS:

Reviewer #1 (Remarks to the Author):

The revised manuscript by Bernal et al. is much improved compared with the original submission. The authors have addressed all of the points raised in the review, and now provide sufficient experimental detail for this approach to be replicated and applied in various experimental systems. The standardization of the nomenclature used to describe the PACK constructs is appropriate.

The authors point out correctly that a PAC-only construct can be used as the "optimal" experimental control for PAC/K-expressing animals. I suggest that such a construct be made available (on Addgene or directly from the authors) so that it can be used by labs wishing to make use of the PAC/K technology.

Minor point: in page 9, the sentence "with further reduced dark activity" appears twice.

We want to thank the reviewer for his supportive comments on the manuscript. The bPAC-only construct will be made available on Addgene, along with the other DNA constructs used in this study. The word duplication has been corrected in the manuscript.

Reviewer #2 (Remarks to the Author):

The authors have done a nice job addressing my comments - except for the rationale behind the design of the split and fused PAC-SthK systems, which they only addressed in the reviewer response document. They should add a rationale around lines 108-110 of the main text. Pending this addition, I support publication of the manuscript in Nature Communications.

Thanks for the positive evaluation of the manuscript. As suggested, we have added the following sentence in the beginning of the results section: "The design of the split system aimed to exclude any impairment of SthK function by the attachment of the PAC, while the fused system should achieve very short cAMP diffusion distances between enzyme and channel."